# Molecular insight into the initial hydration of tricalcium aluminate

Xing Ming [1], Wen Si[2], Qinglu Yu[3], Zhaoyang Sun[3], Guotao Qiu[3], Mingli Cao [2], Yunjian Li [1] ✉ & Zongjin Li [1] ✉

Portland cement (PC) is ubiquitously used in construction for centuries, yet the elucidation of its early-age hydration remains a challenge. Understanding the initial hydration progress of tricalcium aluminate ($C_3A$) at molecular scale is thus crucial for tackling this challenge as it exhibits a proclivity for early-stage hydration and plays a pivotal role in structural build-up of cement colloids. Herein, we implement a series of ab-initio calculations to probe the intricate molecular interactions of $C_3A$ during its initial hydration process. The $C_3A$ surface exhibits remarkable chemical activity in promoting water dissociation, which in turn facilitates the gradual desorption of Ca ions through a metal-proton exchange reaction. The dissolution pathways and free energies of these Ca ions follow the ligand-exchange mechanism with multiple sequential reactions to form the ultimate products where Ca ions adopt fivefold or sixfold coordination. Finally, these Ca complexes reprecipitate on the remaining Al-rich layer through the interface-coupled dissolution-reprecipitation mechanism, demonstrating dynamically stable inner-sphere adsorption states. The above results are helpful in unmasking the early-age hydration of PC and advancing the rational design of cement-based materials through the bottom-up approach.

Portland cement (PC) and concrete are the foremost fabricated and extensively utilized man-made materials due to the swift urbanization of global populace[1,2]. The anticipated demands of projected urbanization necessitate the consumption of ~30 billion tonnes/year of concrete[3], thereby contributing to the substantial production of PC (~4 billion tonnes/year) accompanied by ~2.7 billion tonnes/year of carbon dioxide ($CO_2$) emissions[1,4,5] and large consumption of natural resources[1]. To systematically devise strategies for reducing the carbon footprint of cement and optimizing the related performance (gelation, strength and durability), a comprehensive understanding on the early-age hydration reactions of individual cement components is imperative[4,6]. This holds particularly true for tricalcium aluminate ($Ca_3Al_2O_6$, also denoted as $C_3A$ in cement chemistry notation, where C stands for CaO, A for $Al_2O_3$, S for $SiO_2$, and H for $H_2O$) as it exhibits a proclivity for early-stage hydration preceding other cement

components, thus playing a pivotal role in structural build-up of cement colloids[7–10]. Regrettably, numerous unresolved inquiries persist concerning the intricate dissolution and precipitation processes of $C_3A$ and its relevant hydrates, which govern the setting and early hardening of cement[11–15].

The $C_3A$ hydration manifests distinct stages characterized by the dissolution of surface ions and the precipitation of initial calcium-aluminate-hydrates[15]. Owing to its exceptionally high reactivity with water, the polycrystalline $C_3A$ displays heterogeneous dissolution rates, giving rise to individual pits within a remarkably brief span of <0.1 s. The surface topography subsequently undergoes a coupled transformation marked by the formation of etch pits, point defects and dislocations over a few seconds, ultimately resulting in the development of an Al-rich layer[8,15–18]. Controversies persist in elucidating this phenomenon. One perspective proposes incongruent dissolution of

[1]Faculty of Innovation Engineering, Macau University of Science and Technology, Avenida Wai Long, Taipa, Macao SAR, China. [2]School of Civil Engineering, Dalian University of Technology, Dalian, China. [3]Institute of Applied Physics and Materials Engineering, University of Macau, Avenida da Universidade, Taipa, Macao SAR, China. ✉e-mail: liyunjian@must.edu.mo; zjli@must.edu.mo

Ca and Al ions from the $C_3A$ surface, supported by the initial formation of a Ca-depleted outer layer and a continuous decrease in Ca/Al ratio from the $C_3A$ bulk towards its surface following contact with water[7,15,19]. An alternative hypothesis suggests the interface-coupled dissolution-repreciptation (ICDP) mechanism, positing that the dissolution of Ca and Al ions should be congruent initially or at least minimally enriched in Ca before achieving congruency over time. Dissolved Ca ions elevate the pH, enhancing Al solubility and ultimately stabilizing the Ca/Al ratio in the solution until precipitation occurs. Consequently, the pore solution becomes saturated with $Al(OH)_4^-$, and $AH_3$ gel forms on the crystal surface before other Ca-rich hydrates, resulting in high Ca/Al ratios in aqueous solutions (low Ca/Al ratio on the surface)[7,15,18]. However, this explanation is challenged by recent findings showing that increased pH induced by direct addition of NaOH does not significantly enhance Al ion dissolution[20,21]. Additionally, hydrous Al layers containing bound water may exhibit similar behavior to $AH_3$ gel, rendering their identification through thermal analysis[18]. Although nanoscopic X-ray ptychography has offered direct images of "gel-like" areas on $C_3A$ particle surfaces with a few hundred nanometers thickness after exposure to a sulfate solution for 30–90 min[8], further study is required to confirm this observation[7].

The hydration kinetics of $C_3A$ can be substantially modulated by soluble sulfates to mitigate the impractical occurrence of "flash set"[18]. Cubic $C_3A$ rapidly reacts with $CaSO_4$ to form ettringite (AFt) within minutes[15,22], succeeded by an induction period lasting several hours in isothermal calorimetric curves[13,14]. This induction period typically concludes with a renewed hydration heat release, converting AFt into sulfate-bearing $Al_2O_3$-$Fe_2O_3$-mono (AFm) phases[13-15]. Despite extensive studies, questions remain regarding the retarding effect of gypsum, leading to debates centered around two hypotheses: the barrier and adsorption theories. In the barrier theory, AFt or AFm phases formed on the $C_3A$ surface act as diffusion barriers, impeding bustling ion-water exchanges in hydration reactions[8,14,15]. While the barrier theory has elucidated certain aspects of the typical heat release curve during $C_3A$ hydration, a growing consensus suggests its limitations. Recent findings indicate that AFt exhibits a porous structure that is not conducive to impeding the diffusion of interfacial ions and water molecules[8,23]. Instead, the prevailing view posits that ion pairs of $Ca^{2+}$ and $SO_4^{2-}$ adsorb onto the Al-rich layer of the $C_3A$ surface, effectively obstructing active sites and thereby retarding the hydration kinetics. This inhibition primarily operates through dissolution-controlling mechanisms rather than diffusion-controlling mechanisms[8,18,23].

Despite extensive experimental investigations on initial hydration of $C_3A$, a wide array of debates persists regarding its hydration in both pure water and $CaSO_4$-containing solutions[15]. These debates encompass questions pertained to both the dissolution and participation processes inherent to $C_3A$ hydration. As real-time observation of dissolution and participation processes during early-age $C_3A$ hydration is experimentally challenging and the underlying mechanisms are often deduced indirectly from micro- or macroscopic experimental phenomena, the comprehensive elucidation and explanation heavily rely on the molecular-level understanding of surface activity and interface reactions occurring at the $C_3A$/water interface through the atomistical simulations. To this end, density functional theory (DFT)-based static calculations were initially employed to probe the reactive sites within $C_3A$ crystal[24]. It's theoretically demonstrated that O and Ca ions are active sites in $C_3A$ bulk and are susceptible to electrophilic and nucleophilic attacks, respectively. This suggests that preliminary reaction steps between $C_3A$ and water molecules originate from the surface Ca and O ions[24]. However, static DFT calculations encounter challenges in providing comprehensive kinetic and thermodynamic descriptions of the interactions between water and $C_3A$ surface[25,26]. Subsequent force-field classical molecular dynamics (MD) simulations partially addressed this limitation and revealed that the non-bridging O atoms in the six-member rings of $AlO_4$ tetrahedra are more prone to

protonation than the bridging O sites. Consequently, the surrounded Ca ions can readily accommodate the hydroxide ions ($OH^-$) induced by water dissociation, leading to the formation of calcium hydroxide (CH)[27]. Nevertheless, these non-reactive classical MD simulations appear insufficient to capture the bond breakage and formation involved in the initial dissolution process of $C_3A$, along with the corresponding free energy landscapes. Importantly, they often lack the requisite accuracy and reliability to fully elucidate the intricate physicochemical reactions occurring at the solid/aqueous solution interfaces[28,29].

In response to these challenges, we have undertaken a series of advanced ab-initio (specifically the DFT) calculations to delve deeper insights into the molecular-scale kinetics and thermodynamics of the initial hydration of $C_3A$. Leveraging cutting-edge techniques such as ab-initio molecular dynamics (AIMD) and well-tempered metadynamics (WT-MetaD), we can accurately describe the intricate interfacial reactions and dynamics, thereby addressing the limitations of classical MD and static DFT calculations. We have demonstrated that the initial hydration of $C_3A$ commences with significant surface hydroxylation due to the pronounced dissociation of water molecules in the vicinity areas. This phenomenon further facilitates the gradual desorption of Ca ions through a metal-proton exchange reaction (MPER), following the nucleophilic attack by oxygen ions from water molecules ($O_w$). The dissolution pathways and free energies of these Ca ions are ultimately unveiled to follow the ligand-exchange mechanism with multiple sequential reactions, where the ultimate products are Ca ions adopting fivefold and sixfold coordination, with corresponding free energy barriers of 18.76 kJ/mol and 27.52 kJ/mol, respectively. Finally, these Ca complexes precipitate back on to the remaining Al-rich layer through the interface-coupled dissolution-repreciptation (ICDP) mechanism, demonstrating dynamically stable inner-sphere adsorption states. Our findings shed the light on the molecular scale understanding the initial hydration process of $C_3A$ and the calculated kinetics and thermodynamics can contribute valuable insights for accurately modeling cement hydration and thereby effectively predicting and controlling its macroscopic performance.

## Results

### Distribution of surface ions and water molecules

We began the calculations on conducting an extensive AIMD simulation on the $C_3A$/water interface to gain a comprehensive and basic understanding on the dynamic behavior of $C_3A$ surface in contact with water molecules (Fig. 1). The $C_3A$ surface exhibits a strong attraction to water molecules, boosting their rapid dissociation in the assistant of hydrogen bond (HB) networks[30,31]. Consequently, these dissociated and intact water molecules significantly hydroxylate the exposed $C_3A$ surface, leading to the stabilization of their orientation and distribution (Fig. 1a). After further scrutinizing the density distribution of surface ions and water molecules along the z-axis, we can pinpoint the localized areas of water molecules spanning -14–21 Å, which is subsequently defined as the interface region for calculating the density map and charge distribution. Interestingly, some Ca ions slightly dissolve from the surface during this AIMD simulation, resulting in a relatively flat but broader density profile. This behavior contrasts significantly with the distribution of Al ions on the $C_3A$ surface, where the $AlO_4$ six-membered rings serve as the structural framework of $C_3A$ unit cell and remain dynamically stable throughout the simulations. By combinedly analyzing the number densities of $H_w$ and $O_w$, density ratio ($\rho(O_w)/\rho(H_w)$) and atomic excess ($\Delta\rho$, see Supplementary Note 3 for details) of water molecules, we can distinguish five distinct layers of water molecules with varying thicknesses, which is also presented on γ-$Al_2O_3$ and tricalcium silicate ($Ca_3SiO_5$, $C_3S$) surfaces due to their hydrophilic natures[31,32]. A further analysis shows that the first layer (layer I) locates just below the $C_3A$ surface and primarily consists of OH groups (dissociated water molecules). This layer can be defined as the

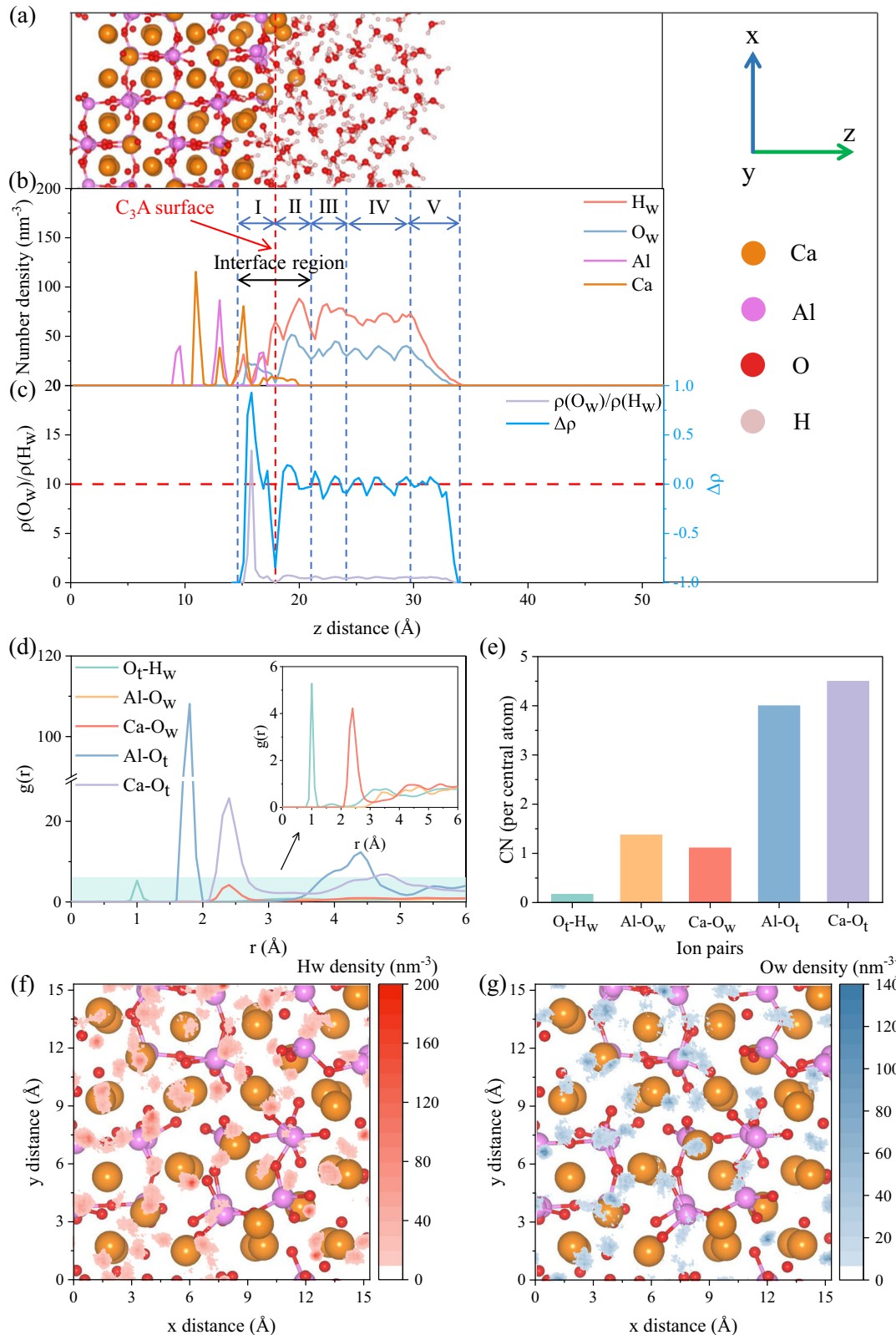

chemisorbed layer of water molecules with a thickness of 4 Å (-14–18 Å), much thicker than those formed on γ-Al$_2$O$_3$ (-2 Å)[31] or C$_3$S surfaces (-1.5 Å)[33]. This indicates that the C$_3$A surface is highly reactive, featuring a greater active site density and a stronger affinity to water molecules. The physisorbed layer (layer II) with a thickness of 3 Å (-18–21 Å) is presented following the first layer where most water molecules are intact but demonstrate reduced movability and higher

structural organization. Consequently, the density peak of O$_w$ slightly precedes that of H$_w$, leading to a small O$_w$-rich region (positive Δρ values and $\rho(O_w)/\rho(H_w) > 0.5$). The third region (-21–24 Å, layer III) occurs with a gentle oscillation both in number density and atomic excess, which is subsequently transferred to the density of 33.33 nm$^{-3}$ and 66.66 nm$^{-3}$ for O$_w$ and H$_w$, respectively (-24–29 Å, layer IV), aligning well with the bulk water density. This indicates that our AIMD

**Fig. 1 | Distribution of surface ions and water molecules. a** Snapshot for the distribution of surface ions and water molecules. **b** Number density of $H_2O$, Ca and Al projected to $z$-axis. **c** Density ratio ($\rho(O_w)/\rho(H_w)$) and atomic excess ($\triangle\rho$) of $H_2O$ projected to $z$-axis. **d** Radial distribution function (RDF), g(r)) of various ion pairs (the shaded region delineates the low coordination of $O_t$-$H_w$, Al-$O_w$ and Ca-$O_w$, and is magnified in the upper right corner) and (**e**) the corresponding coordination numbers (CNs). Number density of (**f**) $H_w$ and (**g**) $O_w$ within the interface region (-14–21 Å) projected to $xy$ plane. For clarity, we denote $O_w$ ($H_w$) as the oxygen (hydrogen) ions of water molecules, and $O_t$ represents the oxygen ions on the $C_3A$ surface. Notably, $O_t$ can be further categorized into bridging and non-bridging oxygen ions ($O_b$ and $O_{nb}$) within the $AlO_4$ six-membered rings. The dynamic distribution of surface ions and water molecules are ensemble-averaged during a -20 ps production run following a -21 ps pre-equilibrium process. The surface line (vertical red dashed line) is determined based on the average positions of the outermost Al ions and used to define the distribution of water molecules in the vicinity regions (layers I – V divided by the vertical bule dashed lines) of the surface in the assistant of density ratio and atomic excess of $H_2O$. The horizontal red dashed line in (**c**) denotes the zero value of atomic excess and serves to delineate the bulk water region (i.e., layer IV).

simulations accurately reproduced the interfacial properties of the $C_3A$/water system. In this context, atomic excess and density ratio fluctuate around 0 and 0.5, respectively, suggesting negligible effects of the $C_3A$ surface on this region. The water/vacuum interface region finally extends from 29 Å to 34 Å (layer V), where the density number and ratio smoothly decrease to 0 as expected. However, unusual negative values of atomic excess in this region are attributed to the ordered water molecules with hydrogen ions orienting towards the vacuum region[31,34]. These findings are further corroborated by an additional classical MD simulation on a larger system based on the ReaxFF force-filed[35,36] (Supplementary Note 4 and Supplementary Fig. 3).

To gain insights into the factors influencing the observed atom distribution, we calculated the radial distribution function (RDF, g(r)) and density maps of $H_w$ and $O_w$ within the interface regions (Fig. 1d–g). The RDF curves reveal clear peaks at -1 Å and -2.40 Å for $O_t$-$H_w$ and Ca-$O_w$, respectively. These results are in good agreement with experimental and theoretical values for O-H and Ca-O distances[37–39], indicating a strong coordination between the water molecules and $C_3A$ surface. Notably, no distinct coordination is observed between Al and $O_w$ ions. This can be attributed to the fact that the coordination state between Al and $O_w$ cannot be adequately sampled within the limited timescale of AIMD simulations. Meanwhile, the stable coordination structure of $AlO_4$ tetrahedra and these $O_t$ ions within the first coordination shell of Al ions may screen direct interactions between Al and $O_w$ ions, thus causing the illusory relatively high coordination numbers (CNs) than those for $O_t$-$H_w$ and Ca-$O_w$ (Fig. 1e actually shows the CNs of Al-$O_w$ in the second coordination shell). These results can be supplemented by classical MD simulations on the RDF of $C_3A$/water interface model, where a low Al-$O_w$ coordination at 1.90 Å with number of 0.23 per/Al atom is observed in the large simulation box with long simulation time (Supplementary Note 4 and Supplementary Fig. 3). This bond length slightly deviates from the equilibrium bond length determined by experiments and simulations[24,27,40], indicating the weak interactions between Al and $O_w$. To further clearly interpret this, the number densities of $H_w$ and $O_w$ in the interface region were projected to $C_3A$ surface (Fig. 1f, g). It's interesting to find the $H_w$ ions prefer to clustering around surface $O_t$ ions, thus hydroxylating the exposed surface. This is consistent with the discussions on atom distribution and RDF. Besides, surface Ca ions are predominantly surrounded by $O_w$ ions, whether in the form of OH groups or intact water molecules. When considering the earlier discussions, this coordination environment facilitates the slight dissolution of Ca ions from the surface through MPER, which successfully interprets the leaching of mental ions from mineral/aqueous solution interfaces[41,42] although such step (superficial protonation) should be ruled out as being rate-determining[16].

## Dynamics of surface ions and water molecules

We next analyzed the interfacial dynamics of $C_3A$/water system (Fig. 2). The hydrogen bonds (HBs) are formed between the surface $O_t$ and $O_w$ ions when the $C_3A$ surface is attacked by water molecules, thus disrupting the HB network and decreasing HB numbers among water molecules before 21 ps (Fig. 2a). Subsequently, the HB numbers of $O_t$-$O_w$ are dynamically stable, leading to a slower fluctuation in HBs

among water molecules after 21 ps. This indicates that our system has reached an equilibrium state, not only in terms of potential energy and system temperature but also with regard to interfacial properties. Consequently, the results are highly reliable and reproducible. By further checking the ion pairs involved in $O_t$-$O_w$ HBs, we find that HBs primarily form between $O_{nb}$ and $O_w$ ions, as evidenced by the two curves displaying similar quantities over time. Interestingly, most of $O_{nb}$ ions, despite being hydroxylated and attached to hydrogen ions from adjacent water molecules, only act as HB acceptors upon closer inspection of the coordination details. As a result, we hardly observe any proton hops between the surface $O_t$ and $O_w$ ions, which should be commonly present in mineral/aqueous solution interfaces and facilitated by continuous HB networks[30,43]. This limited proton hopping is yet attributed to the highly nucleophilic nature of the $O_t$ ions, particularly the $O_{nb}$ ions, which form strong bonds with $H_w$ ions and consequently restrict proton transfers. This behavior is similar with what has been observed on ionic oxygen ions on $C_3S$ surfaces[24,27,43].

We also quantified the bond number density for all possible bonds between $C_3A$ surface and water molecules (Fig. 2b). Specifically, the $O_w$-$H_w$ bond number density represents the number density of OH groups and is equivalent to the numbers of dissociated water molecules. The evolution of bond number densities can be divided into three distinct stages, except for Al-$O_w$, which remains relatively stable. In the first stage (-0–9 ps), the bond number densities increase rapidly over time, particularly pronounced for Ca-$O_w$ due to the high density of active sites on $C_3A$ surface. Subsequently, these active sites are gradually saturated by either hydrogen ions or OH groups, thus slowing down increase rate in bond number density from 9 ps to 21 ps. Finally, the bond number density reaches a steady state as the surface active sites become nearly saturated with water molecules (-21–41 ps). The similar responses are also observed on $C_3S$ and dicalcium silicate ($Ca_2SiO_4$, $C_2S$) surfaces[44–46], although they demonstrate lower bond number densities than those of $C_3A$ (Fig. 2c), meaning the diffusion of water molecules (including the protons and OH groups) are the main controlling factor for the very early-age hydration process (or strictly defined as the wetting) of $C_3A$[14]. However, the bond number density of Al-$O_w$ shows no significant increase during the entire simulation, consisting with the RDF results discussed earlier (Fig. 1d). This implies that Al ions are much less prone to dissolution from $C_3A$ surface in the assistance of water molecules compared to Ca ions as they demonstrate distinct coordination environments and electronic nature[15]. Consequently, we can speculate that Ca and Al ions follow incongruent dissolution pathways, leading to the formation of a Ca-leaching layer during the initial hydration process of $C_3A$. This hypothesis can be experimentally verified by the evolutions of surface zeta potentials and compositions during $C_3A$ hydration[7,8,19,23].

The time-dependent radial distribution function (TDRDF) was also calculated to verify the dynamic evolution of these interfacial properties (Fig. 2d–f). Consistent with the bond number density results, the TDRDF intensities for $O_t$-$H_w$ (Fig. 2d) and Ca-$O_w$ (Fig. 2f) exhibit an increase with time, eventually reaching stable values after 21 ps. Conversely, the TDRDF intensity for Al-$O_w$ remains almost unchanged with a relatively low value during the whole simulation. We also calculated the time-dependent density (TDD) profiles of surface ions and water

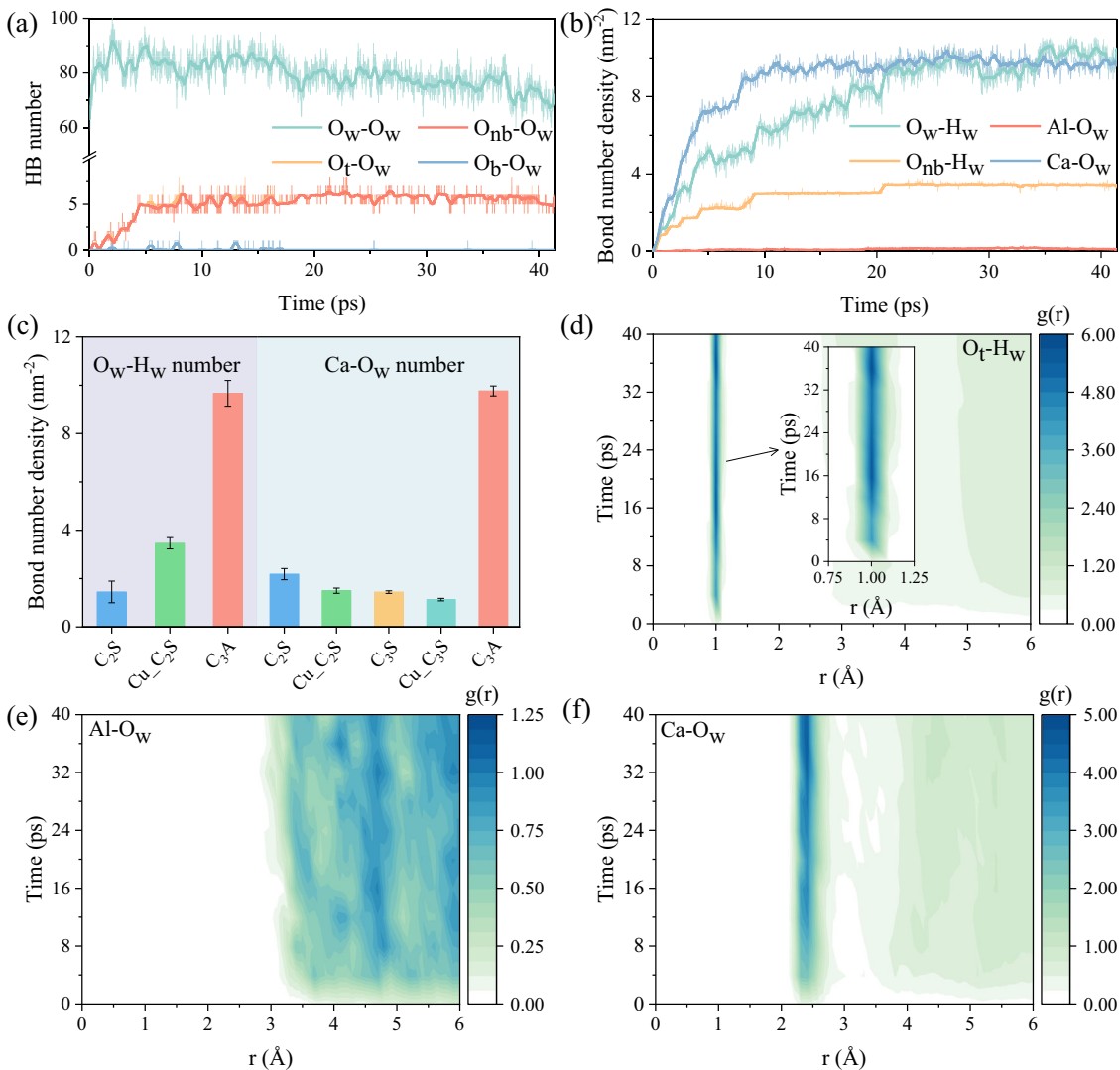

**Fig. 2 | Dynamics of surface ions and water molecules. a** Hydrogen bond (HB) number and **b** bond number density evolution during the whole simulation. **c** Comparison of bond number density of various cement components. The raw data for calcium silicates is from references[44,45]. Time-dependent radial distribution function (TDRDF) of (**d**) $O_t$-$H_w$, (**e**) Al-$O_w$ and (**f**) Ca-$O_w$, respectively. The shaded lines with spikes in (**a**) and (**b**) depict the original curves of HB number and bond number density, respectively. For clarity and comparison, these curves have been averaged (illustrated as colored solid lines). To assess the bond number density across various surfaces of cement components, averaging is conducted on their equilibrium values with error bars denoting the standard deviation. The TDRDF profiles are computed by averaging RDF data over 1 ps trajectories sampled every 4 ps.

molecules (Supplementary Note 5 and Supplementary Fig. 4). The surface $O_t$ and Ca ions are the most active sites in $C_3A$ surface, accommodating the protons and water oxygens, respectively, when in contact with water molecules. Thus, a narrow distribution of $H_w$ and $O_w$ ions are finally observed with the time increase, while a broader distribution of Ca ions is presented in the later stage (Supplementary Fig. 4). Compared with the rapid re-arrangement of surface $O_t$ and Ca ions, the surface Al ions remain stable due to the thermodynamic stable six-membered ring structure, thereby resulting in a spatial separation with three layers of distribution (Supplementary Fig. 4c). Consequently, it is plausible that a higher energy barrier may be required for dissolving Al ions from the perfect $C_3A$ surface that of Ca ions.

## Electronic properties
Based on the above discussions on distribution and dynamics of surface ions and water molecules involved in $C_3A$/water interface, we next investigated the insightful interaction mechanisms between water molecules and surface active sites, particularly the $O_t$ and Ca ions to

reveal their reactivity nature at electron scale (Fig. 3 and Supplementary Note 1). The analysis of charge density difference ($\Delta q$) reveals a significant redistribution of charges in the interface region during the binding process (Fig. 3a, b). This redistribution underscores that high affinity of $C_3A$ surface to water molecules is primarily driven by electrostatic interactions between the active sites and water. Similar charge redistribution can also be presented on $C_3S$/water interface[43]. Specifically, the charge accumulates (positive $\Delta q$) around the $O_w$ and $O_t$ ions, while delocalizes (negative $\Delta q$) predominantly around the Ca and $H_w$ ions, which mirrors the bond formation and dynamic evolution previously described. To gain deeper insights, we also computed the total and local density of states (TDOS and LDOS) for different surface ions and water molecules (Fig. 3c, d). It is evident that the $O_a$ ions (comprising $O_b$, $O_{nb}$ and $O_w$ ions) dominate the valence-band maximum (VBM), which can be further distinguished with $O_w$ ions exhibiting the closest energy level to Fermi level ($E_f$) in the valence-band. Conversely, Ca ions primarily contribute to the conduction-band minimum (CBM) with the closest energy level to $E_f$ in the conduction-band. These findings suggest that $O_w$ ions predominantly serve as

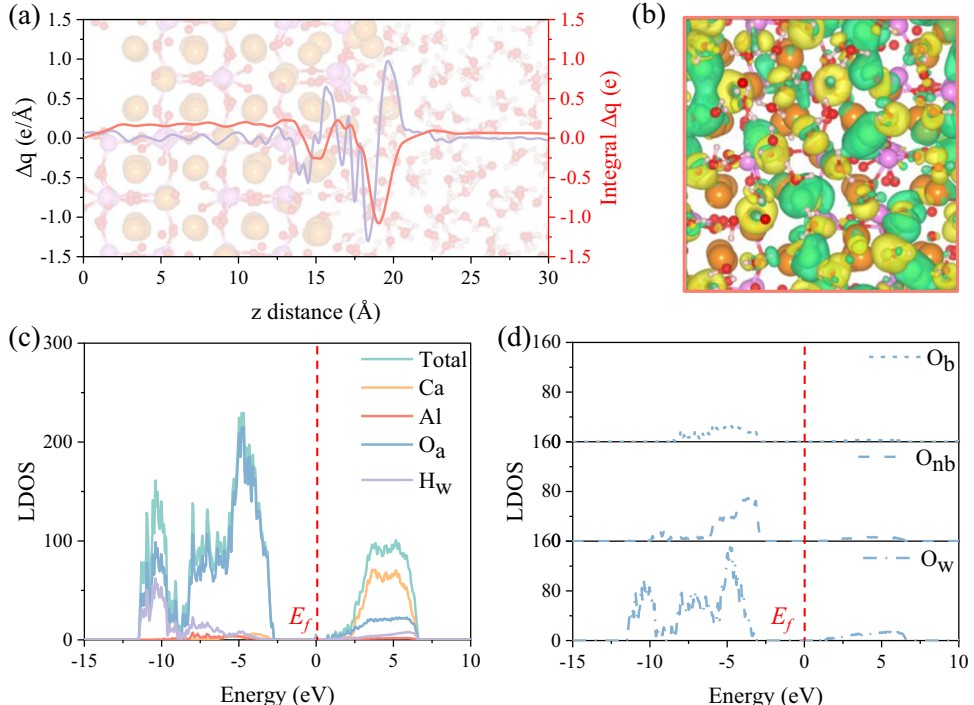

**Fig. 3 | Electronic properties of C₃A/water interface.** Charge distribution of interface region projected to (**a**) *z*-axis and (**b**) xy plane (iso-surface value of $2.62 \times 10^{-2}$ e/Å³). **c** Local density of states (LDOS) of various surface ions and (**d**) detailed LDOS of various oxygen ions. The positive $\triangle q$ value (**a**) or yellow areas (**b**) refer charge accumulation regions around these ions, while a negative value or green areas mean charge depletion regions.

basic sites in water molecules, susceptible to electrophilic attacks, while Ca ions act as acid sites on C₃A surface, subject to nucleophilic attacks[24,47]. This corresponds well with DFT studies on identifying the active sites of C₃A, both with and without interaction with $CaSO_4$[24,48].

## Calcium dissolution with CN (Ca-$O_w$) from 1 to 3

To date, we have uncovered the insightful interaction process between C₃A surface and aqueous solutions at molecular scale and determined the active sites and the underlying electronic nature governing these reactive encounters, which finally induces slight dissolution of surface Ca ions in the assistant of water molecules. Nevertheless, it should be noted that these slightly dissolved Ca ions are still strongly interacted with the surface $O_t$ ions and perform an inner-sphere complex, meaning that the full dissolution of Ca ions from C₃A surface is still rare event during the limited AIMD sampling time scale. Here, we revealed the full dissolution pathways and free energy landscapes for two kinds of surface Ca ions with different coordination numbers (CNs) in the assistant of WT-MetaD. This choice is substantiated by the existing literatures, which indicates that the coordination environments of central ions play a pivotal role in controlling their sequential dissolution pathways and free energies[32,49]. In the following discussions, we primarily focus on the low-coordinated Ca ion, while the high-coordinated state with water molecules can be found in the Supplementary Note 7 and Supplementary Fig. 8.

For such dissolution, we can readily distinguish four distinct free energy minima on the first-stage free energy surface (FES) for Ca dissolution, corresponding to the initial, intermediate and final states involved in this first-stage reaction (Fig. 4a). Specifically, the selected Ca ion initially adopts a metastable state of (3,0) on the freshly cleaved (001) surface due to the disruption of its coordination environment. Subsequently, the surface undergoes significant reconstruction, transforming the (3,0) state into a (4,0) state through the rotation and slight displacement of AlO₄ tetrahedra (Supplementary Note 10 and Supplementary Movie 1). The (4,0) state quickly transfers to A (4,1) state when water molecules encounter the C₃A surface, indicating the

water molecules can stabilize the slightly detached surface Ca ions. Upon crossing a small free energy barrier ($\Delta A^{\ddagger}$ (A-B) = 1.88 kJ/mol), a lower free energy basin B (4,2) emerges on the FES, representing the most stable sate among all the four minima states (Fig. 4b). Through further inspecting the configuration evolution (Fig. 4c), we can find two additional water molecules (the water molecules include both the intact and dissociated ones as our reaction coordinate cannot well-define the OH groups and intact water molecules) and four surface $O_t$ ions coordinate with the selected Ca ions, forming an octahedral structure. This coordination environment is typically associated with the stability of Ca ions, observed both in aqueous solutions and cement hydrates[37,50]. The reaction pathways then diverge into two paths at state B and converge at state D (3,3) with a well-defined octahedral coordination environment consisting of three surface $O_t$ ions and three $O_w$ ions (Fig. 4b, c). For path A-B-C-D, it follows a ligand exchange reaction mechanism[32,49,51], where the bond breakage between Ca and $O_t$ occurs by crossing two free energy barriers ($\Delta A^{\ddagger}$ (B-C) = 14.47 kJ/mol and $\Delta A^{\ddagger}$ (C-D) = 3.07 kJ/mol) and an intermediate state C (4,3) only when additional water molecules or OH groups encounter the Ca ion. This reaction mechanism is also observed in simulating the dissolution of Ca ions from the C₂S and C₃S surfaces[32,52,53], where the seven-coordinated structure is a crucial state for the subsequent reactions between water molecules and Ca ions, despite its severe distortion into a octahedral-like coordination[54]. For path A-B-D, the bond breakage of Ca-$O_t$ and bond formation of Ca-$O_w$ occur simultaneously, thereby only crossing one free energy barrier ($\Delta A^{\ddagger}$ (B-D) = 18.05 kJ/mol) to form the stable six-coordinated configuration for Ca ion. The rate-determining step for the later sequential reactions has a relatively higher free energy barrier than the first one, indicating path A-B-C-D is more kinetically favorable with the free energy barrier for rate-determining step of 14.47 kJ/mol.

## Calcium dissolution with CN (Ca-$O_w$) from 4 to 6

In the first-stage dissolution of Ca ions, the FES covers coordination numbers with water molecules <4. Crossing a large free energy barrier

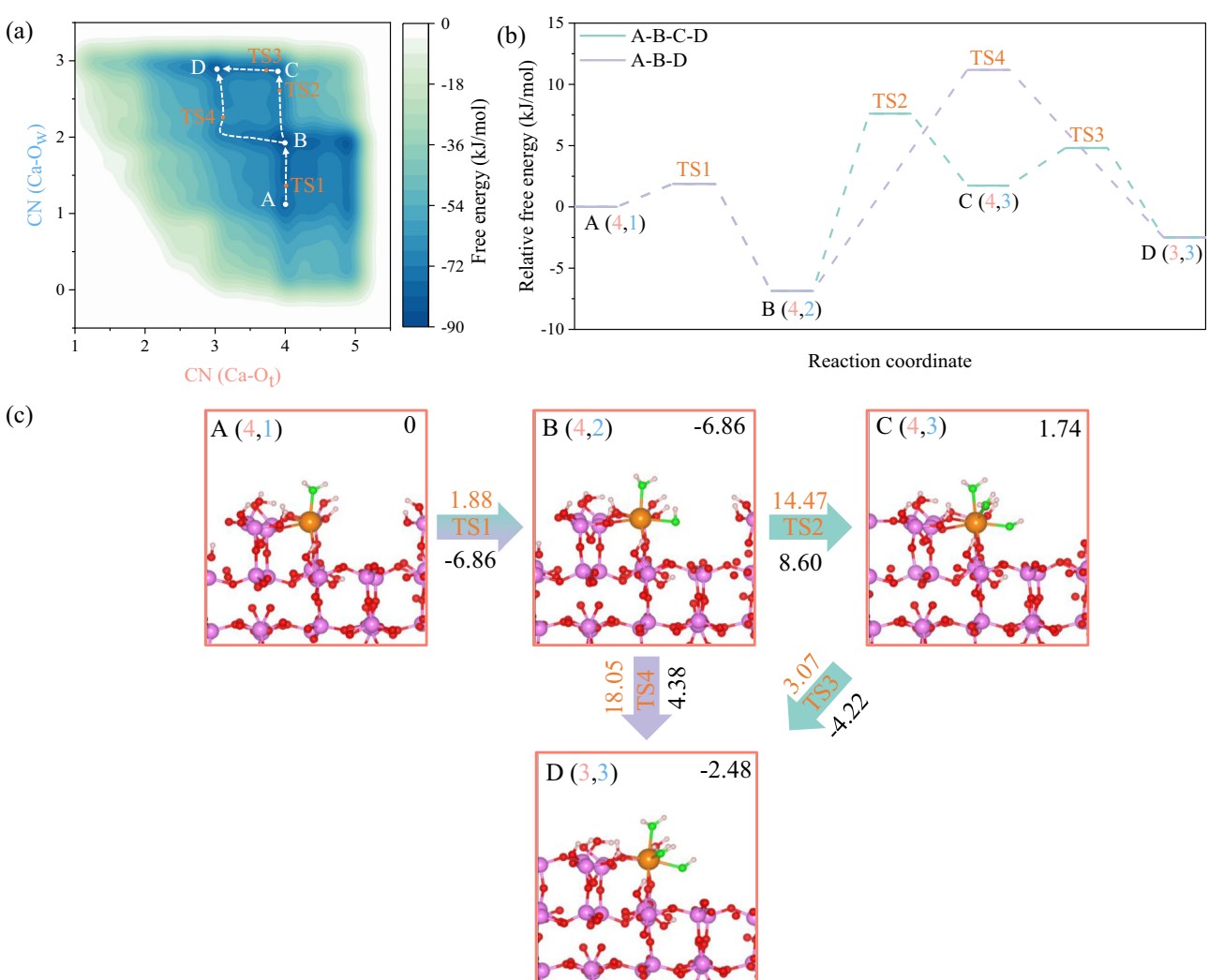

**Fig. 4 | Calcium dissolution with coordination numbers to water molecules (CN (Ca-O$_w$)) ranging from 1 to 3. a** Free energy surface, **b** reaction coordinate and (**c**) corresponding snapshots of configuration evolution along the reaction pathways. In the above representations, the sates along the dissolution pathways are notated in the form of X (CN (Ca-O$_t$), CN (Ca-O$_w$)), where X indicates the state number on the free energy surface or the snapshots (e.g., the A, B, etc.), CN (Ca-O$_t$) and CN (Ca-O$_w$) represent coordination numbers of Ca to O$_t$ and O$_w$, respectively in state X.

"TS" denotes the transition state. The upper right corners display the states with Helmholtz free energy values (in kJ/mol) relative to state A. The saffron yellow values above the arrows represent free energy barriers (in kJ/mol), while the black values below the arrows denote the free energy differences between two adjacent states (in kJ/mol). The arrows with varied colors signify the distinct reaction pathways that align with those depicted in (**b**).

is necessary to further dissolve the Ca ions when accommodating more water molecules and breaking the surface interactions with these Ca ions. Therefore, we introduced a "wall" to restrict the collective variables (CVs) from fully exploring the regions of interest (where CN (Ca-O$_w$) is >3). A more complex FES is then unmasked with six distinguished free energy minima (Fig. 5a). The reaction pathways start from the stable state E (1,4) and easily and quickly transfer to a more stable state F (1,5) with a small free energy barrier of 4.13 kJ/mol (Fig. 5b, c). Although this six-coordinated structure demonstrates similar configuration with those in the first-stage dissolution (B and D), it's not the most stable state in such FES. Instead, it can further transform to the most stable state I (0,5) with a free energy barrier of 18.76 kJ/mol following the ligand-exchange mechanisms, despite such configuration demonstrates heavily distorted trigonal bipyramid (D$_{3h}$) structure[32,49]. Meanwhile, we also note a kinetically unfavorable reaction pathway for transferring state F to state I through two intermediate states G (1,6) and H (0,6) after crossing three free energy barriers (ΔA$^‡$ (F-G) = 33.52 kJ/mol, ΔA$^‡$ (G-H) = 27.52 kJ/mol and ΔA$^‡$ (H-I) = 56.32 kJ/mol). It is interesting to note that sixfold coordination of

Ca ion is not the most stable structure on such FES, although it has comparative relative free energies with I (0,5) (Fig. 5c). This indicates that state I and H can transform mutually through liberating or accommodating one water molecule both from the thermodynamic and kinetic views. Such transformation can also occur between state I and state J (0,4). However, the reverse reaction, involving accommodating one more water molecule, is more thermodynamically and kinetically favorable when compared to the forward and backward reaction barriers (ΔA$^‡$ (I-J) = 33.27 kJ/mol and ΔA$^‡$ (J-I) = 2.13 kJ/mol). By further inspecting the configuration evolution (Fig. 5c), we finally conclude that one of the surface O$_{nb}$ ions near the dissolved Ca ion has strong electrostatic interactions with the central Ca ion coordinated by five or six water molecules, which can then partially destroy the hydration shell of the central Ca ion by further forming the distorted fivefold coordination for such Ca ion. On the other hand, this interaction can also perturbate the coordination shell of Ca ion with an octahedral configuration and even transform it to sevenfold or sixfold coordination states (i.e., state G (1,6) or F (1,5)), which can be further confirmed by our equilibrium AIMD simulation implemented on the

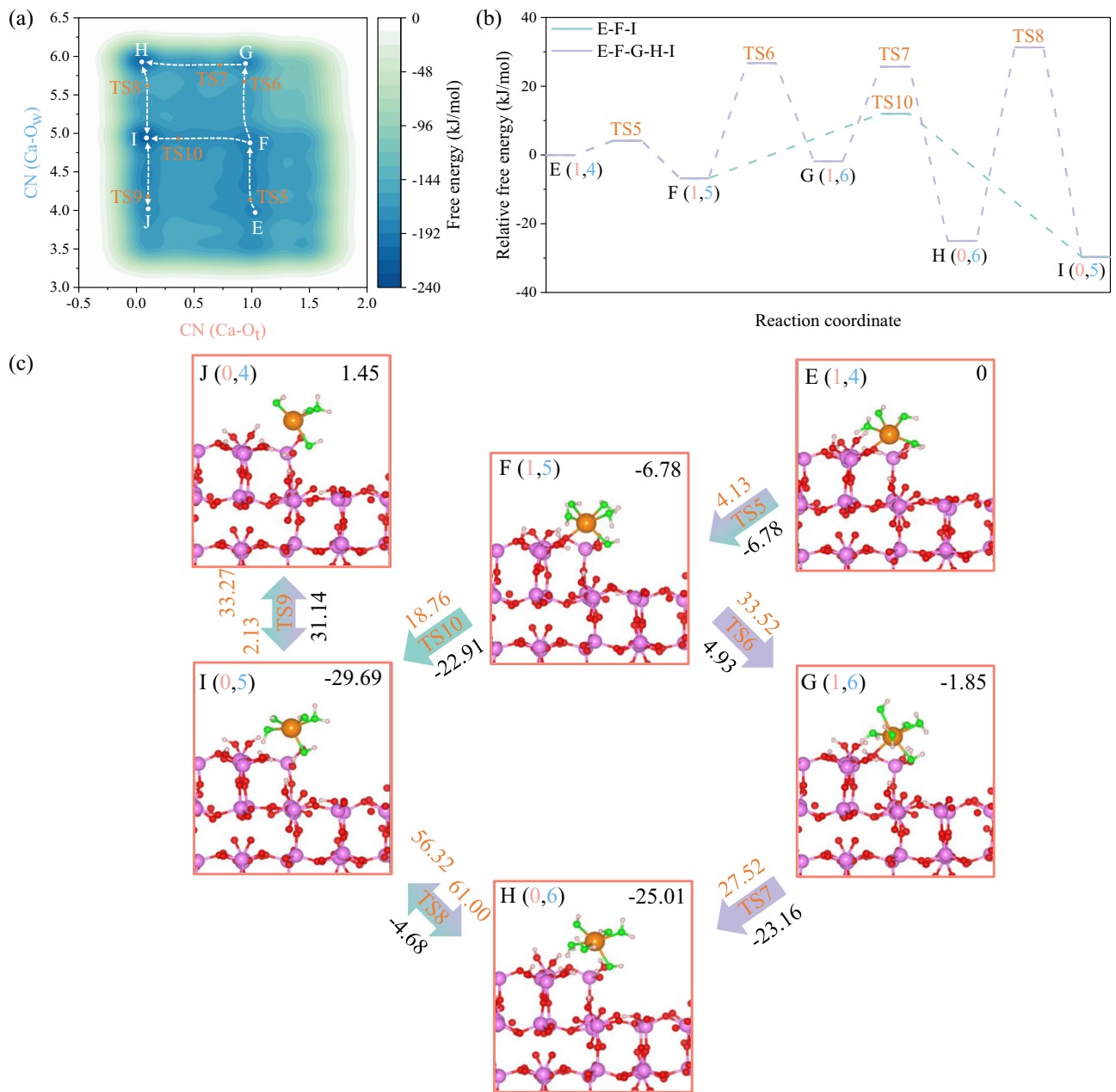

**Fig. 5 | Calcium dissolution with coordination numbers to water molecules (CN (Ca-O$_w$)) ranging from 4 to 6. a** Free energy surface, **b** reaction coordinate and **c** corresponding snapshots of configuration evolution along the reaction pathways. In the above representations, the sates along the dissolution pathways are notated in the form of X (CN (Ca-O$_t$), CN (Ca-O$_w$)), where X indicates the state number on the free energy surface or the snapshots (e.g., the A, B, etc.), CN (Ca-O$_t$) and CN (Ca-O$_w$) represent coordination numbers of Ca to O$_t$ and O$_w$, respectively in state X. "TS" denotes the transition state. The upper right corners display the states with Helmholtz free energy values (in kJ/mol) relative to state E. The saffron yellow values above the arrows represent free energy barriers (in kJ/mol), while the black values below the arrows denote the free energy differences between two adjacent states (in kJ/mol). The arrows with varied colors signify the distinct reaction pathways that align with those depicted in (**b**).

state H (0,6) (Supplementary Note 8 and Supplementary Fig. 9). This unique phenomenon is consistent with the simulation statements that the hydration structure of Ca$^{2+}$ is highly variable in aqueous solutions with several shallow local minima in the free-energy curves[55].

## Discussion
### Molecular-level C$_3$A hydration
We can now provide a detailed description of the initial hydration processes of cubic C$_3$A at molecular scale based on the above ab-initio calculations (Fig. 6). The C$_3$A surface exhibits a pronounced chemical affinity to water molecules, leading to a rapid and substantial dissociation of these molecules. The H$_w$ and O$_w$ then strongly coordinate

with the surface O$_{nb}$ and Ca ions to initiate the dissolution of surface Ca ions through the MPER[41,42]. Subsequent reactions of water dissociation and surface hydroxylation heterogeneously and incongruently desorb the surface Ca ions prior to Al ions, promoting the formation of various inner-sphere complexes (Ca ions with different CNs to water). The following full dissolution of the selected Ca ions presents a complex FES and reaction coordinate due to the intricate interactions between ligand water molecules and surface active ions. However, it generally follows ligand-exchange mechanisms[32,49,51]. When considering state I (0,5) as the final state, the minimum free energy pathway (MFEP) consists of E-F-I with a free energy barrier of 18.76 kJ/mol for the rate-determining step (from F to I). On the other hand, when state H (0,6) is

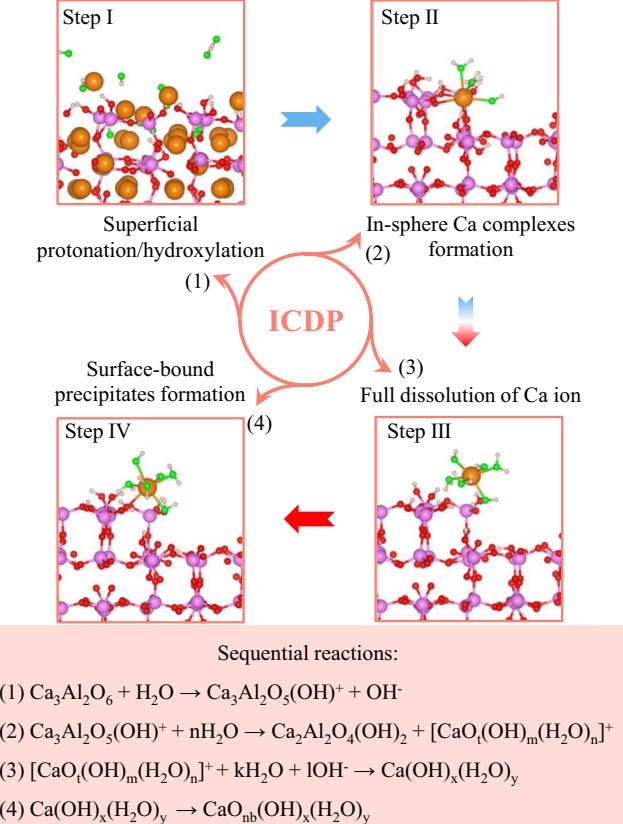

**Sequential reactions:**

(1) $Ca_3Al_2O_6 + H_2O \rightarrow Ca_3Al_2O_5(OH)^+ + OH^-$

(2) $Ca_3Al_2O_5(OH)^+ + nH_2O \rightarrow Ca_2Al_2O_4(OH)_2 + [CaO_t(OH)_m(H_2O)_n]^+$

(3) $[CaO_t(OH)_m(H_2O)_n]^+ + kH_2O + lOH^- \rightarrow Ca(OH)_x(H_2O)_y$

(4) $Ca(OH)_x(H_2O)_y \rightarrow CaO_{nb}(OH)_x(H_2O)_y$

**Fig. 6 | Molecular-level C₃A hydration and the associated sequential reactions.** For clarity, the Ca-coordinated O ions in water molecules and hydroxy groups are highlighted in green, while the other Ca ions (excluding the selected one for the dissolution reaction) are omitted to clearly illustrate the coordination state of the formed Ca complex from step II to IV.

taken as the final state, the MFEP involves E-F-G-H with a higher free energy barrier of 27.52 kJ/mol for the rate-determining step (from G to H). Nevertheless, both states can dynamically convert to surface-bound coordination states (i.e., state G (1,6) or F (1,5)) according to our following equilibrium AIMD simulation, indicating such Ca complexes can adsorb and precipitate on the hydrous Al-rich layers and yielding the positively charged C₃A surface (Supplementary Note 9 and Supplementary Fig. 10). This phenomenon indirectly supports the experimentally observed positive zeta potentials of C₃A[14,19,23,56] and further confirms that Ca ions may inhibit C₃A dissolution at high concentrations[19,23]. Finally, the surface-bound precipitates $(CaO_{nb}(OH)_x(H_2O)_y)$ containing several liberated OH groups will form an amorphous CH layer on the Ca leaching surface. Such amorphous layer is kinetically favored as the local coordination environment is enriched with OH groups and benefits the full hydration of Ca ions, despite the adjacent aqueous solution is significantly undersaturated with respect to even crystalline CH[16]. Moreover, this amorphous layer can also act as the precursor and finally transform to AFt or various AFm phase induced by the thermodynamic driving force[15]. The overall reactions recognize the formation of surface alteration layers (SALs) consisting of Al-rich layer and re-participated Ca complexes during the initial hydration processes of C₃A induced by the ICDP mechanism[57] and emphasize the critical roles of incongruent dissolution-triggered Al-rich layers in determining surface activity and interface reactions[18,23]. When SO₄²⁻ ions are also included in the electrical double layer (EDL) of the partially dissolved C₃A surface, a pronounced inhibition on C₃A dissolution is realized[23]. This effect is achieved by adsorbing SO₄²⁻ ions on the surface through electrostatic attraction with the surface-bound Ca complexes, thereby obstructing the coordination sites that would be otherwise occupied by H⁺ or OH⁻ ions that are known to catalyze the dissolution of alumina[58] and C₃S[32].

## Implications for cement hydration

The molecular-level descriptions of the initial hydration of C₃A can be seamlessly integrated into real-life cement hydration to optimize its performance through a bottom-up approach, especially when combined with multi-scale simulation and experimental methods. First, we have systematically identified the elementary reaction steps and rate-determining steps for Ca dissolution. The rate equation for such dissolution can be precisely formulated using the transition-state theory[59], advancing the power function-based empirical rate equation that describes the overall dissolution process of C₃A in the absence of certain knowledge about the reaction sequences and the rate-determining steps[60]. Second, the ab-initio calculations can serve as a benchmark for force-field parameters specifically designed for investigating the initial hydration of C₃A through classical MD simulations. This enables us to explore the hydration process of C₃A under conditions closer to actual scenarios, considering factors such as surface defects, solid solution, pH, etc. Finally, the revealed kinetics and thermodynamics can be integrated into the thermodynamic database of cubic C₃A. These data are typically experimentally acquired through a dissolution approach, where the water activity and flow rate are specially designed to facilitate in-situ observation of surface topography evolution[7,16,17,61] while minimizing the precipitation of secondary or intermediate solids. In this way, the dissolution reaction is proceeded in the forward direction and the rate equation can be written as a power function of reactant concentrations or activities[16]. By combining these three aspects, we can comprehensively capture the full hydration processes of C₃A from the molecular scale to the macro-scale and interpret the early-age hydration of cement in real-life.

In conclusion, we have identified the sequential reactions involved in the C₃A/water interface through atomistic simulations and provided a comprehensive description of the molecular-level initial hydration of C₃A based on the simulations and pertinent literatures. This study paves the way for understanding and controlling the early-age hydration reactions of cement components, with significant implications for the design and optimization of cement-based materials. A deeper comprehension of the initial hydration of C₃A can facilitate the design of admixtures to regulate and modify kinetics and mechanisms, thereby manipulating the rheology, structural build-up and even durability of cement. However, it is essential to acknowledge the complexity of actual cement hydration reactions, particularly in the case of C₃A, which involves multi-step and interrelated reactions affected by crystal polymorphs, solution chemistry, surface defects, etc. Future research should consider these variables and strive for a more detailed and quantitative description of Al dissolution and the early precipitation of calcium-aluminate-hydrates.

## Methods

### Model construction

The cubic crystal (Pa-3 space group) structure of C₃A utilized in this study is sourced from reference[62], with a sizable unit cell measuring 15.26 × 15.26 × 15.26 Å³. Initially, the bulk C₃A structure was optimized using the Vienna ab-initio Software Package (VASP)[63,64] (Supplementary Note 1). Subsequently, to create the solid/aqueous solution interface model, the original bulk model was cleaved along the (001) plane (Supplementary Note 2). The stoichiometric lower part of the C₃A bulk was then fixed (bulk area) and only the upper part was relaxed (surface area) during the following simulations[31,43]. A ~ 15 Å layer containing 122 water molecules was then added to this surface to achieve an aqueous environment with a density of 1 g/cm³ and establish the C₃A/water interface. The vacuum space between the interface model

and its periodic image was ~21 Å. As a result, the final cell dimensions were $15.39 \times 15.39 \times 52.13$ Å$^3$, comprising a total of 630 atoms.

## AIMD simulation

The AIMD simulations were conducted using the CP2K/Quickstep package[65] by employing density functional theory (DFT) based on a hybrid Gaussian plane wave (GPW) approach[66]. In this scheme, orbitals are represented using an atom-centered Gaussian-type basis set, while an auxiliary plane wave basis set is used to expand the electron density in reciprocal space. The convergence criteria for energy and self-consistent field (SCF) were set to $10^{-12}$ Ha and $10^{-6}$ Ha, respectively. To enhance wave function optimization and SCF convergence, an orbital transformation (OT) method[67] was employed, along with a wave function extrapolation strategy known as always stable predictor corrector (ASPC). In the calculations, the valence electrons considered were the 1 s electron for H, 2 s and 2p electrons for O, 3 s and 3p electrons for Al, and 3 s, 3p, and 4 s electrons for Ca. The remaining core electrons were described using Goedecker-Teter-Hutter (GTH) pseudopotentials[68,69]. A double-ζ Valence Polarized (DZVP) Gaussian-type basis set was employed[70], and the energy cutoff for the auxiliary plane wave basis set was set to 400 Ry. The exchange-correlation effects were treated with the Perdew-Burke-Ernzerhof (PBE) functional[71], including dispersion correction using the Grimme D3 method[72]. Due to the large size of the supercell, only the Γ point was used for Brillouin-zone integrations in all calculations. To capture the structural evolution of the interface model and bulk solution, Born-Oppenheimer molecular dynamics (BOMD) was utilized under canonical ensemble (NVT) conditions. A Nose-Hoover thermostat maintained the temperature at 300 K. Deuterium masses were used for protons to minimize nuclear vibrational frequencies with a timestep of 1 fs, which is a commonly used value in the literature to prevent energy drifts and maintain computational stability[43,73]. The AIMD simulation was divided into two stages: an initial ~21 ps equilibrium stage and a final ~20 ps stage for statistical analysis of the distribution and dynamics of surface ions and water molecules (Supplementary Fig. 2). Additionally, the last frame of this AIMD run was specifically extracted for calculations of electronic properties using the DFT-implemented VASP package[63,64], along with the VASPKIT plugin[74]. All atomistic configurations were visualized and analyzed using the VESTA software[75].

## WT-MetaD simulation

The dissolution process and the associated free energies of Ca ions from the C$_3$A surface cannot be directly observed during the timescale of MD simulations, even though C$_3$A exhibits exceptionally high reactivity and dissolves within 0.1 s[15]. Such rapid events are still rare in conventional MD sampling. Therefore, to address this limitation, we employed an enhanced sampling method known as metadynamics to sample the dissolution pathways and generate corresponding free energy surfaces within the limited timescale of AIMD simulations. All WT-MetaD simulations concerning Ca dissolution were conducted using the CP2K software[65] plugin integrated with the PLUMED package[76,77]. We employed two CVs characterized by the CNs to construct the two-dimensional free energy surfaces for Ca dissolution. Specifically, the CNs (Ca-O$_{t/w}$) represented the coordination numbers of the selected surface Ca ion with all the oxygen ions from the C$_3$A slab (O$_t$) and from water molecules (O$_w$), respectively. These CNs were defined according to the rules implemented in PLUMED[76,77]

$$CN(Ca - O_{t/w}) = \sum_{i \in Ca} \sum_{j \in O_{t/w}} s_{ij} = \sum_{i \in Ca} \sum_{j \in O_{t/w}} \frac{1 - \left(\frac{r_{ij}-d_0}{r_0}\right)^n}{1 - \left(\frac{r_{ij}-d_0}{r_0}\right)^m} \quad (1)$$

where $r_{ij}$ is the distance between atom $i$ and $j$. $d_0$ is the function central value. $s_{ij}$ is a rational type of switching function describing the coordination between atom $i$ and $j$. $r_0$ is the acceptance distance of the

switching function. $n$ and $m$ are two integer exponents with $n < m$. Specific parameter values were employed: $d_0$ and $r_0$ were set at 2.36 Å and 0.4 Å, respectively, based on the radial distribution function of Ca-O$_{t/w}$. The values of $n$ and $m$ were chosen as 4 and 10, respectively. Gaussian hills were added every 20 steps for both CVs with an initial height of 3.5 kJ/mol and a width of 0.15, following established methodology[32,76,78]. A bias factor of 10 was applied for Ca dissolution. Furthermore, quadratic walls with a force constant of 2000 kJ/mol were intentionally positioned in the WT-MetaD simulations to confine the fluctuations of the CVs within the defined area of interest. Details regarding the time evolution of both CNs, convergence tests for free energy surfaces, and the evaluation of errors between two free energy minima can be found in the Supplementary Note 6.

## Data availability

The data supporting the pivotal findings of this study are comprehensively documented within the article and Supplementary Information file. For access to the raw data produced during the current study, interested parties may request them directly from the corresponding authors.

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

## Acknowledgements

The authors gratefully acknowledge the support from the Science and Technology Development Fund (FDCT), Macao SAR (0147/2022/A3 and 0096/2023/AFJ) [Z.L.]. Additionally, the Beijing Super Cloud Center (BSCC, http://www.blsc.cn/) and Bohrium Cloud Platform of DP Technology (https://bohrium.dp.tech/) are acknowledged for providing HPC resources that have significantly contributed to the research outcomes presented in this paper. The first author (Xing Ming) wishes to express appreciation for the support and fruitful discussions provided by Prof. Yongqing Cai from the University of Macau in conducting DFT calculations on electronic properties by utilizing VASP.

## Author contributions

X.M., Y.L., and Z.L. conceptualized the research. X.M. conducted the DFT calculations, analyzed the findings, and drafted the manuscript. Y.L. and Z.L. conceived and supervised the project. X.M., W.S., Q.Y., Z.S., G.Q., and M.C. deliberated on the context and importance of conducting this simulation work. X.M., M.C., and Z.L. revised the manuscript prior to submission. All authors participated in result discussions and manuscript contributions.

## Competing interests

The authors declare no competing interests.
