## [Peer Review File · Nature Communications]

Molecular insight into the initial hydration of tricalcium aluminateReviewer #1 (Remarks to the Author):

The research paper relies heavily on computational simulations (AIMD and WT-MetaD) to understand the initial hydration progress of tricalcium aluminate (C3 A). However, there is a significant concern about the reliability and accuracy of these simulations as they often make simplifying assumptions and approximations that may not accurately represent real-world conditions.

The paper lacks a critical assessment of the limitations and assumptions made in the computational simulations. Understanding the accuracy and precision of these simulations is crucial for the validity of the results and their applicability to real-life cement hydration processes.

The paper emphasizes the catalytic activity of the C3 A surface in promoting water dissociation, but it fails to provide a comprehensive comparison with existing experimental data. Without such comparison, the significance and applicability of the findings remain uncertain.

The conclusions drawn from the simulations are presented with a high degree of confidence, without adequately discussing the uncertainties associated with the computational methods and assumptions. A more cautious and nuanced interpretation of the results is essential for a rigorous scientific discussion.

The paper does not discuss potential experimental verification or validation of the simulation results. Without experimental confirmation, the findings remain speculative and lack real-world validation.

The language used in the paper could be more concise and precise. The excessive use of technical jargon and complex sentence structures makes it difficult for a broader audience to understand the research findings.

The paper does not adequately address alternative theories or models that may explain the observed phenomena. Considering and discussing alternative explanations would strengthen the robustness of the research.

The research could benefit from a more thorough discussion of practical implications and applications of the findings. How can this understanding of the initial hydration process of C3 A be practically utilized to optimize cement performance? This aspect is not sufficiently addressed.

The introduction of the paper could be improved to provide a clearer context and motivation for the research. It should clearly articulate the research gap or problem that the study aims to address and why understanding the initial hydration process of C3 A is crucial.

Reviewer #2 (Remarks to the Author):

The hydration mechanisms of cement component have always been a focus in cement research community. Even with years of research, these mechanisms, especially at molecular scale, are still not fully understood. This study is very interesting and it moves the field forward. The authors gave a full dynamic description on the initial hydration process of C3A at molecular scale and uncovered the mechanisms behind. I think the paper presents a clear and well-corroborated message that is relevant for understanding the key step involved in the initial hydration process of C3A. Here are some comments and suggestions for authors to consider:

Comment 1

The retarding effect of gypsum was well elaborated in the introduction section. However, this study focused on the initial hydration process of C3A. The logical relationship between these two problems was not well clarified. Please add more descriptions in this section to make it clearer.

Comment 2

It is generally recognized that the dissolution of tricalcium silicate (a major component of cement clinker) is anisotropic due to the crystal defects, e.g., dislocations and point defect. Similarly, C3A is a kind of crystal material that probably contains crystal defects. The particle surface with crystal

defects may contain higher lattice energy compared with other sites on the surface. This implies that the particle surface is not stable at some sites, which probably leads to the anisotropic dissolution of C3A. Did the authors consider the effect of crystal defects in the simulation? if not, could authors explain why these effects were not included? The information regarding the anisotropic dissolution of tricalcium silicate, please refer to [CBM 265(2020) 120458], [CCR,156(2022)106763], [CBM,249(2020) 118535].

Comment 3

In Supplementary materials: "During the ~41 ps AIMD sampling, temperature and potential energy evolution profiles are presented in Fig. S1." Please double-check the figure cited in this sentence.

Comment 4

Please check subscript of C3A (e.g., Page 3, line 88).

Comment 5

In References : The authors need to maintain consistency in their writing. Therefore, please check subscript of Ca₃Al₂O₆ (e.g., Page 23, line 578), please check the style of reference [20] (Page 24, line 628) and reference [25] (Page 25, line 642).

Comment 6

In Computational methods: How many computational resources were utilized in the WT-MetaD and AIMD simulations and how many time would it take for the two simulations? This information should be added in the manuscript to help readers to reproduce the whole process.

Comment 7

In Model construction: The authors mentioned that the original bulk model was cleaved along the (001) plane (on line 162). why did authors choose (001) face to establish solid/aqueous interface model? The reason for doing this needs to be clarified in the manuscript.

Reviewer #3 (Remarks to the Author):

The authors performed a series of ab-initio calculations to understand the initial hydration of tricalcium aluminate. My suggestions and comments are as follows:

1. The authors have mentioned about some catalytic effect. I do not see any. A catalyst decreases the barrier for a reaction between two compounds and gets regenerated after the reaction. If I understand it correctly, the C3A surface is one of the reactants in this case. Ca ions get released from the surface, but the protons stay at the surface.
2. It has been mentioned that the addition of 122 water molecules provides an aqueous environment with a density of 1 g/cm³ in the C3a/water interface. This would be impossible to achieve due to an additional vacuum space between the interface model. Authors should check their model to ensure that the accuracy of their simulations is reliable and accurate.
3. The authors should refer below papers:
<https://doi.org/10.1016/B978-0-12-404504-0.50021-6>
<https://doi.org/10.1063/1.2768063>
4. The authors should suggest chemical reaction schemes for the initial hydration of tricalcium aluminate.
5. In the SI information, authors have mentioned about the well-established force field models for exploring interactions between water molecules and cement components [12-15]. Authors should provide set of force field parameters used in these calculations with validated molecular models of C3A. I could not find validated C3A force field parameters in the references mentioned above.
6. Comparison between ab-initio and all-atom force field model would make sense only when the force field parameters were specifically developed for C3A mineral and validated with experimental data.
7. A minor comment is the necessity to improve the language and grammatical structure of the manuscript.

Responses to Reviewers' Comments

Reviewers' comments:

Reviewer #1 (Remarks to the Author):

The research paper relies heavily on computational simulations (AIMD and WT-MetaD) to understand the initial hydration progress of tricalcium aluminate (C_3A). However, there is a significant concern about the reliability and accuracy of these simulations as they often make simplifying assumptions and approximations that may not accurately represent real-world conditions.

Reply: We strongly disagree with the reviewer's one-sided critique of the ab-initio calculations used in our research. In computational methods, such as density functional theory (DFT) based molecular dynamics, static calculations, and metadynamics, there are inherent approximations, including the Born-Oppenheimer nonrelativistic and independent electron approximations, along with the Hohenberg-Kohn theorems I and II. These approximations simplify solving the time-dependent Schrödinger equation for a many-body system into a ground-state total energy calculation for a non-interacting many-body system using the variational principle. DFT-based calculations have been successfully employed to study and predict physical and chemical properties of various materials in solid, liquid, gas, and mixed phases, demonstrating high reliability and accuracy. This differs from classical molecular dynamics, which heavily relies on parameterized force fields and is limited to specific simulation systems. In computational models, the interface model is commonly used to investigate complex physicochemical reactions, such as catalytic, dissolution, and participation processes involved in solid/liquid interfaces [1-4]. The underlying mechanisms and molecular interactions revealed by ab-initio calculations have been supported and confirmed by advanced characterization methods, reinforcing the reliability and accuracy of studying complex physicochemical reactions using the interface model. Regarding simulation conditions, temperature and pressure in our simulations are consistent with real-world conditions due to advancements in metadynamics, allowing us to sample rare events, like calcium dissolution, at normal temperature and pressure. This is different from the annealing method, which uses high temperatures to improve reaction probabilities not

observed at normal conditions. Additionally, we remark that, in contrast with most of the calculations, no constraint is applied in our protocol, such as to guide the reaction along a given pathway, namely neither the reactant nor the product are set up in prearranged configurations. The successful use of this method can be found in various literature references [4-7] and the basic principles can be found in our supplementary materials and the cited references. In summary, we believe that the ab-initio calculations in our research are highly reliable and accurate for simulating and predicting physicochemical reactions involving C₃A/water interface. This confidence is based on the theoretical foundation of our computational methods and the suitability of our models.

The paper lacks a critical assessment of the limitations and assumptions made in the computational simulations. Understanding the accuracy and precision of these simulations is crucial for the validity of the results and their applicability to real-life cement hydration processes.

Reply: In the preceding discussions, we underscored the reliance of DFT-based calculations on inherent approximations. Thus, these calculations yield highly accurate and reliable results although their validation necessitates specialized experimental methods such as in-situ electrochemical shell-isolated nanoparticle-enhanced Raman spectroscopy (SHINERS), extended X-ray absorption fine structure (EXAFS), or dynamic nuclear polarization-enhanced solid-state nuclear magnetic resonance (NMR) [3, 8-11]. A potential limitation of our research stems from the computational model, which exclusively considers the intricate interactions between the perfect C₃A surface and water. This simplification may deviate from the actual cement hydration processes, where complex reactions involving multiple phases contribute to the overall cement hydration reaction. However, it is noteworthy that comprehensive hydration models for cement cannot be fully incorporated into molecular-scale computational models at present due to the absence of reliable models encompassing all intricate reactions in cement hydration and the impractical computational demands associated with such models. Consequently, the exploration of real-life cement hydration using all-atom models remains unfeasible. Our focus remains on elucidating the fundamental reaction principles of C₃A with water, a pursuit unattainable through experimental means. This focus is particularly significant for comprehending the broader context of cement

hydration processes. The simulation of the initial hydration process of C_3A employs a C_3A /water interface, representative of a typical solid-liquid reaction model in actual cement hydration processes. This model is accurately and reliably captured through recent advancements in ab-initio molecular dynamics (AIMD) accompanied by metadynamics simulations. Consequently, our approach facilitates the observation of the molecular-scale dynamics of the initial hydration reaction of C_3A . The simulation data obtained can be further applied in thermodynamic modeling of real-life cement hydration, exemplifying a potential application of ab-initio calculations in understanding practical cement hydration scenarios, which is also anticipated by a recent experimental literature [12].

The paper emphasizes the catalytic activity of the C_3A surface in promoting water dissociation, but it fails to provide a comprehensive comparison with existing experimental data. Without such comparison, the significance and applicability of the findings remain uncertain.

Reply: The discovery of the crucial roles of C_3A surface in facilitating water dissociation constitutes a novel and pivotal aspect of our research. Furthermore, the indispensable roles played by this catalytic activity in subsequent reaction steps of C_3A with water underscore the advantages and necessity of the computational approach over traditional experimental methods. To the best of the author's knowledge, there is currently no comparable experimental data that provides such profound insights into the catalytic activity of the C_3A surface toward water molecules. In line with the typical theoretical work, our study assumes a critical role in offering a predictive and forward-looking perspective for achieving precise control over C_3A hydration. While direct comparisons between our DFT calculations and experiments may be challenging due to their independence and distinct focuses on different scales, it is essential to note that the predictability and autonomy of ab-initio calculations contribute to their unique strengths. The absence of direct comparisons does not undermine the significance of our findings, as ab-initio calculations often serve as guiding principles for experimental endeavors. Moreover, while no direct comparable experimental data currently exists, the inclusion of indirect experimental data in our research serves to validate our simulation findings. This added dimension further supports the significance and applicability of our simulation results, reinforcing their importance even in the absence

of directly comparable experimental data at present. On the other hand, recent developments in in-situ Raman spectroscopy have successfully elucidated the structure and dissociation of interfacial water on metal and mineral surfaces [3, 8, 13]. This avenue appears promising for uncovering the intricate reactions between C_3A surface and water, and further research in this direction should be pursued.

The conclusions drawn from the simulations are presented with a high degree of confidence, without adequately discussing the uncertainties associated with the computational methods and assumptions. A more cautious and nuanced interpretation of the results is essential for a rigorous scientific discussion.

Reply: As emphasized earlier, our computational methods and models, albeit relying on inherent approximations, are meticulously tailored to investigate the intricate interfacial reactions between water and the C_3A surface, ensuring the reliability of our conclusions. Each derived inference stems from a cohesive set of mutually reinforcing calculations. For instance, the conclusion regarding the pronounced affinity of C_3A surface for water molecules is substantiated by analyses of density distribution, radial distribution function (RDF), and the dynamics of interfacial waters. Additionally, we ascertain the nature of C_3A surface affinity through the calculation of charge distribution in the interface region. Consequently, the conclusions reached are unequivocally reliable, accurate, and subject to thorough deliberation. Also, we have revised our conclusions by incorporating more relevant experimental and simulation evidence to deepen our discussions on the initial hydration of C_3A (see section 3 in the revised manuscript) and we further provide a pathway for implicating our results in real-life cement hydration. These discussions finally provide a cautious and nuanced interpretation of our findings.

The paper does not discuss potential experimental verification or validation of the simulation results. Without experimental confirmation, the findings remain speculative and lack real-world validation.

Reply: We have explicitly mentioned one of the key motivations for our research: “molecular-level understanding of surface activity and interface reactions occurring at the C_3A /water interface.” This motivation is based on the fact that “real-time observation of dissolution, nucleation, and participation processes during early-age C_3A

hydration is experimentally challenging and the underlying mechanisms are often deduced indirectly from micro- or macroscopic experimental phenomena." Similar to other theoretical work that often serves as a guide for experiments, our study provides a predictive and forward-looking perspective on achieving precise control over C_3A hydration. As of the manuscript submission, we have not come across any experimental papers related to molecular-scale observations of C_3A hydration. Despite advanced atomic-resolution scanning transmission electron microscopy (STEM), time-resolved resonant anomalous X-ray reflectivity (TRAXR) measurements and quick scanning extended X-ray absorption fine structure (QEXAFS) are recently used to investigate the ion-exchange reaction and rapid precipitate formation at the mineral/water interfaces [14-16], their applicability in probing the initial hydration processes of C_3A crystals- from surface hydroxylation to the dissolution of surface Ca ions-remains uncertain. Furthermore, these in-situ methods often necessitate specialized sample treatments or alterations to the testing environment, introducing disparities with conditions prevalent in both real-life cement hydration process and molecular-scale simulations. Consequently, direct correlation between these experimental data and simulation results are still challenging. In the manuscript, we briefly compared our findings with calculation results from other researchers and inferred certain outcomes based on our calculations to indirectly align with existing microscopic experimental observations. However, the development of suitable techniques and methods to capture the rapid reactions of C_3A with water at the molecular scale remains a formidable challenge in this field. We eagerly anticipate forthcoming experimental work, which is technically feasible, to validate our theoretical predictions.

The language used in the paper could be more concise and precise. The excessive use of technical jargon and complex sentence structures makes it difficult for a broader audience to understand the research findings.

Reply: We have revised relevant sentences in response to the reviewer's comments, focusing on enhancing conciseness, precision, and clarity (see the highlight sentences in the revised manuscript).

The paper does not adequately address alternative theories or models that may explain the observed phenomena. Considering and discussing alternative explanations would

strengthen the robustness of the research.

Reply: We have explicitly referenced certain theories and models to interpret observed phenomena throughout the simulation process. Specifically, the metal-proton exchange reaction (MPER) model is employed to elucidate the pronounced affinity of C₃A surface for water molecules, along with the subsequent slight detachment of surface Ca ions. In detailing the complete dissolution process of surface Ca ions, we rely on the ligand-exchange mechanism to provide a clear understanding of the dissolution landscape. These established theories and models enable us to trace the full dissolution pathways of surface Ca ions at molecular scale with accuracy and reliability. However, in response to the reviewer's concerns, we have supplemented the revised manuscript with additional relevant theories or models to bolster the robustness of our research. For instance, we employed the interface-coupled dissolution-precipitation (ICDP) mechanism to explain the formation of surface-bound precipitates, which further provides a deeper insight into the origins of the experimentally observed positive zeta potentials of C₃A and the retarding effect of sulfates (see section 3.1 in the revised manuscript).

The research could benefit from a more thorough discussion of practical implications and applications of the findings. How can this understanding of the initial hydration process of C₃A be practically utilized to optimize cement performance? This aspect is not sufficiently addressed.

Reply: We appreciate the reviewer's valuable input regarding the translation of simulation findings into practical applications for cement. In response, we have addressed the pertinent issues and expanded upon the potential applications of our findings, outlining recommendations for optimizing cement performance through a bottom-up approach, as detailed below and also included in the revised manuscript:

Implications for cement hydration. The molecular-level descriptions of the initial hydration of C₃A can be seamlessly integrated into real-life cement hydration to optimize its performance through a bottom-up approach, especially when combined with multi-scale simulation and experimental methods (Fig. R1). First, we have systematically identified the elementary reaction steps and rate-determining steps for Ca dissolution. The rate equation for such dissolution can be precisely formulated using the transition-state theory [17], advancing the power function-based empirical rate

equation that describes the overall dissolution process of C₃A in the absence of certain knowledge about the reaction sequences and the rate-determining steps [18]. Second, the ab-initio calculations can serve as a benchmark for force-field parameters specifically designed for investigating the initial hydration of C₃A through classical MD simulations. This enables us to explore the hydration process of C₃A under conditions closer to actual scenarios, considering factors such as surface defects, solid solution, pH, etc. Finally, the revealed kinetics and thermodynamics can be integrated into the thermodynamic database of cubic C₃A. These data are typically experimentally acquired through a dissolution approach, where the water activity and flow rate are specially designed to facilitate in-situ observation of surface topography evolution [19-22] while minimizing the precipitation of secondary or intermediate solids. In this way, the dissolution reaction is proceeded in the forward direction and the rate equation can be written as a power function of reactant concentrations or activities [21]. By combining these three aspects, we can comprehensively capture the full hydration processes of C₃A from the molecular scale to the macro-scale and interpret the early-age hydration of cement in real-life.

Fig. R1. Schematic diagram of the initial hydration of C₃A and implications for cement hydration. The experimental image is from ref. [23].

The introduction of the paper could be improved to provide a clearer context and motivation for the research. It should clearly articulate the research gap or problem that the study aims to address and why understanding the initial hydration process of C₃A is crucial.

Reply: We have effectively addressed the aforementioned issues and have revised the introduction in accordance with the provided comments. Furthermore, we have placed emphasis on the motivation and novelty of our research, particularly in investigating the initial hydration of C₃A (see the introduction in the revised manuscript).

Reviewer #2 (Remarks to the Author):

The hydration mechanisms of cement component have always been a focus in cement research community. Even with years of research, these mechanisms, especially at molecular scale, are still not fully understood. This study is very interesting and it moves the field forward. The authors gave a full dynamic description on the initial hydration process of C₃A at molecular scale and uncovered the mechanisms behind. I think the paper presents a clear and well-corroborated message that is relevant for understanding the key step involved in the initial hydration process of C₃A. Here are some comments and suggestions for authors to consider:

Reply: We express our gratitude to the reviewer's meticulous examination and insightful comments on our work. We appreciate his/her positive remarks, noting that our work is deemed "very interesting" and commending it for advancing the field. Additionally, we acknowledge the recognition of our presentation as "clear and well-corroborated," with relevance in understanding the key step involved in the initial hydration process of C₃A, which aligns precisely with the central motivation and significance behind simulating the initial hydration process of C₃A in our research. Below, we provide a point-to-point response to the reviewer's comments, and the relevant revisions are highlighted in red in the revised manuscript.

Comment 1

The retarding effect of gypsum was well elaborated in the introduction section. However, this study focused on the initial hydration process of C_3A . The logical relationship between these two problems was not well clarified. Please add more descriptions in this section to make it clearer.

Reply: We have diligently addressed the raised issues in the revised manuscript. Additionally, we have reorganized the introduction section to establish a clear and logical relationship between the initial hydration process of C_3A and the retarding effect of gypsum (see the introduction section in the revised manuscript).

Comment 2

It is generally recognized that the dissolution of tricalcium silicate (a major component of cement clinker) is anisotropic due to the crystal defects, e.g., dislocations and point defect. Similarly, C_3A is a kind of crystal material that probably contains crystal defects. The particle surface with crystal defects may contain higher lattice energy compared with other sites on the surface. This implies that the particle surface is not stable at some sites, which probably leads to the anisotropic dissolution of C_3A . Did the authors consider the effect of crystal defects in the simulation? if not, could authors explain why these effects were not included? The information regarding the anisotropic dissolution of tricalcium silicate, please refer to [CBM 265(2020) 120458], [CCR,156(2022)106763], [CBM,249(2020) 118535].

Reply: The dissolution response of C_3A polycrystals is inherently anisotropic, primarily attributed to the substantial presence of defect sites (dislocations, point defects, grain boundaries, etc.) induced by high-temperature calcination and the supercooling processes [21, 22, 24]. Additionally, single crystal C_3A exhibits diverse reactive sites and atoms, susceptible to chemical attack, owing to distinct electronic properties and local coordination environments [25]. In both cases, defect sites on the mineral surface exhibit higher dissolution kinetics than perfect sites, primarily due to the presence of low-coordinated surface ions [7, 26]. In our simulations, we do not explicitly account for the effects of crystal defects on dissolution dynamics for two main reasons: (1) The perfect C_3A bulk inherently contains Ca ions with diverse coordination environments, resulting in varying reactivity to water molecules [25]. After cleaving on (001) surface and contacting with water molecules, the coordination environments of

surface Ca ions are greatly altered due to the surface reconstruction. This leads to distinct coordination numbers of Ca ions and their disparate dissolution dynamics. As illustrated in our study, by considering two representative surface Ca ions after AIMD relaxation (one with low coordination to water molecules in the main manuscript and the other with high coordination to water molecules in the supplementary material S7), we observe varying dissolution landscapes and free energy surfaces due to their inherently different coordination environments. Consequently, we extrapolate that surface defects would similarly influence the dissolution pathways of Ca ions, as these defects induce comparable effects on the coordination environments of surface Ca ions through the surface reconstruction process. (2) Another reason for excluding defects from our interface model stems from the fact that C_3A components typically manifest as polycrystals in real cement. Surface etch pits and grain boundaries are critical factors in determining the dissolution rates of C_3A [21, 22]. However, incorporating all-atom models with etch pits or grain boundaries into typical AIMD and metadynamics simulations is computationally challenging due to limitations in computational power. Additionally, determining precise and reliable defect concentrations and distributions on C_3A surface proves challenging due to variations in calcination procedures, raw material compositions, polymorphism of C_3A crystals, and the lack of high-precision measurement methods. Although the recent publication paves the way for determining the crystal defects of alite at atomic scale through the high-resolution TEM [27], the specific distribution and concentration of such defects are still lack. Even if accurate defect concentrations and distributions were available, interpreting them in computational models for AIMD and metadynamics would be impractical due to the relatively small model sizes leading to unreasonably high concentrations and densities of surface defects, potentially deviating simulation results from reality. Thus, we opt for a perfect C_3A (001) surface to construct our interface model for probing the initial hydration process, which is both technically feasible and theoretically reasonable.

Comment 3

In Supplementary materials: “During the ~41 ps AIMD sampling, temperature and potential energy evolution profiles are presented in Fig. S1.” Please double-check the figure cited in this sentence.

Reply: The figure referred to should be Fig. S2 in the supplementary materials. We

have rectified the error in citation.

Comment 4

Please check subscript of C₃A (e.g., Page 3, line 88).

Reply: We have thoroughly reviewed the entire manuscript and corrected the pertinent subscript errors.

Comment 5

In References: The authors need to maintain consistency in their writing. Therefore, please check subscript of Ca₃Al₂O₆ (e.g., Page 23, line 578), please check the style of reference [20] (Page 24, line 628) and reference [25] (Page 25, line 642).

Reply: We have carefully reviewed the entire manuscript and ensured consistency in the writing style of the reference types.

Comment 6

In Computational methods: How many computational resources were utilized in the WT-MetaD and AIMD simulations and how many time would it take for the two simulations? This information should be added in the manuscript to help readers to reproduce the whole process.

Reply: The WT-MetaD and AIMD simulations were conducted on high-performance computing clusters at the Beijing Super Cloud Computing Center, utilizing 64 CPUs of the Intel(R) Xeon(R) CPU E5-2678 v3@2.50 GHz type. This information has been added to supplementary materials and is also provided in detail below:

Table R1. Simulation details for all AIMD and WT-MetaD simulations.

Simulation project	Timestep (fs)	Simulation time (ps)	Elapsed time
AIMD simulation on C ₃ A/water interface	1	41.38	
Ca dissolution with CN (Ca-Ow) from 1 to 3 (low coordination to water molecules)	1	120.33	
Ca dissolution with CN (Ca-Ow) from 4 to 6 (low coordination to water molecules)	1	87.13	~ 2000 steps/day with 64 CPUs
Ca dissolution with high CN (Ca-Ow)	1	78.09	
Equilibrium AIMD simulation for state E	0.5	8.36	

Comment 7

In Model construction: The authors mentioned that the original bulk model was cleaved along the (001) plane (on line 162). why did authors choose (001) face to establish solid/aqueous interface model? The reason for doing this needs to be clarified in the manuscript.

Reply: The unit cell of bulk C₃A model employed in our simulations belongs to the cubic system, resulting in the existence of the following independent low-index surfaces in symmetry: (001), (011) and (111). We thus computed their surface energies using the provided eq. (R1) and determined that the (001) surface possesses the lowest surface energy, indicating its preference for cleavage and exposure to water molecules. The calculations on surface energies have been included in the supplementary materials S2 and Table S2 to elucidate the rationale behind selecting this surface for the interface model.

$$\gamma_s = \frac{1}{2A} (E_s^{unrelax} - NE_b) + \frac{1}{2A} (E_s^{relax} - E_s^{unrelax}) \quad (R1)$$

Table R2. Surface energies of different cleaved surfaces.

Serials	Structure	Surface energy (J/m ²)
(001)		1.91
(011)		2.23

(111)

2.03

Reviewer #3 (Remarks to the Author):

The authors performed a series of ab-initio calculations to understand the initial hydration of tricalcium aluminate. My suggestions and comments are as follows:

Reply: We would like to thank the reviewer for the careful examination and review on our work. Below we would like to make a point-to-point reply. Besides, the relevant revisions are also highlighted in red in the revised manuscript.

1. The authors have mentioned about some catalytic effect. I do not see any. A catalyst decreases the barrier for a reaction between two compounds and gets regenerated after the reaction. If I understand it correctly, the C_3A surface is one of the reactants in this case. Ca ions get released from the surface, but the protons stay at the surface.

Reply: In the bulk water area, the water molecules remain intact during ~41 ps AIMD simulation. In contrast, in the interface region, water molecules experienced significant dissociation into protons and hydroxy groups, subsequently binding to surface O ions and Ca ions, respectively. Upon comparing the distinct behaviors of water molecules in these two regions, we concluded that the C_3A surface can catalyze the dissociation of water molecules. Despite undergoing substantial reconstruction and a slight detachment of Ca ions from the surface during this reaction, the C_3A surface generally maintains its crystal skeleton, and the detached Ca ions remain in the inner sphere adsorption state. As a result, it can be loosely considered that the C_3A surface regenerates after the dissociation reaction of water molecules. However, it's essential to acknowledge that this description deviates slightly from a strict definition of catalytic roles. We have revised the relevant statements in the manuscript to ensure clarity and accuracy (see lines 25-28).

2. It has been mentioned that the addition of 122 water molecules provides an aqueous environment with a density of 1 g/cm^3 in the C_3A /water interface. This would be impossible to achieve due to an additional vacuum space between the interface model. Authors should check their model to ensure that the accuracy of their simulations is reliable and accurate.

Reply: The reviewer has expressed significant concerns regarding the realization of a water density of 1 g/cm^3 in our simulation box, attributed to the presence of an additional vacuum layer. We believe there may be some misunderstanding in the reviewer's interpretation of our model construction, as outlined in our manuscript. As a result, we have included additional descriptions of model construction to enhance clarity and comprehension (see lines 497-499). The achievement of a water density of 1 g/cm^3 is attained through the initial distribution of 122 water molecules in a $15.39 \times 15.39 \times 15.39 \text{ \AA}^3$ box. This water box is then placed on the C_3A (001) surface, with an additional $\sim 20 \text{ \AA}$ vacuum layer positioned above this water layer (further details can be found in the supplementary material S1). Importantly, the thickness of the vacuum layer is sufficient to isolate the influence of the periodic image of the C_3A surface on the water layer. Our results indicate that the C_3A surface affects the water distribution only within a $\sim 10 \text{ \AA}$ thickness (as shown in Fig. 1 in the revised manuscript). It's worth noting that during subsequent AIMD simulations, the water molecules relax and penetrate into the C_3A surface, which slightly increases the thickness of the vacuum layer from the initial $\sim 20 \text{ \AA}$ to a final $\sim 21 \text{ \AA}$. However, the bulk water, with a density of 1 g/cm^3 , is maintained with a thickness of $\sim 5 \text{ \AA}$ throughout the entire simulation period (see Fig. S4 in the supplementary materials). This underscores the strong affinity of the C_3A surface for water molecules, a characteristic observed in other mineral/water interfaces as referenced in [28, 29]. Furthermore, it's worth noting that similar methods for constructing mineral/water interface models have been employed in numerous studies [3, 7, 29].

3. The authors should refer below papers:

<https://doi.org/10.1016/B978-0-12-404504-0.50021-6>

<https://doi.org/10.1063/1.2768063>

Reply: We express our gratitude to the reviewer for providing the relevant key literatures. Following a thorough examination of the literatures, we have engaged in a

more comprehensive discussion of our simulation data. This deeper exploration aims to provide additional experimental context for our simulations (see lines 403-406, 430-432 and 446-449).

4. The authors should suggest chemical reaction schemes for the initial hydration of tricalcium aluminate.

Reply: We have successfully addressed the aforementioned issues and supplemented the chemical reaction schemes and descriptions for the initial hydration of C₃A based on the AIMD and WT-MetaD simulations (see Fig. S11 in the supplementary materials and also Fig. R1 below).

Molecular-level C₃A hydration. We can now provide a detailed description of the initial hydration processes of cubic C₃A at molecular scale based on the above ab-initio calculations (Fig. R1). The C₃A surface exhibits a pronounced chemical affinity to water molecules, leading to a rapid and substantial dissociation of these molecules. The H_w and O_w then strongly coordinate with the surface O_{nb} and Ca ions to initiate the dissolution of surface Ca ions through the MPER [30, 31]. Subsequent reactions of water dissociation and surface hydroxylation heterogeneously and incongruently desorb the surface Ca ions prior to Al ions, promoting the formation of various inner-sphere complexes (Ca ions with different CNs to water). The following full dissolution of the selected Ca ions presents a complex FES and reaction coordinate due to the intricate interactions between ligand water molecules and surface active ions. However, it generally follows ligand-exchange mechanisms [5, 7, 32]. When considering state I (0,5) as the final state, the minimum free energy pathway (MFEP) consists of E-F-I with a free energy barrier of 18.76 kJ/mol for the rate-determining step (from F to I). On the other hand, when state H (0,6) is taken as the final state, the MFEP involves E-F-G-H with a higher free energy barrier of 27.52 kJ/mol for the rate-determining step (from G to H). Nevertheless, both states can dynamically convert to surface-bound coordination states (i.e., state G (1,6) or F (1,5)) according to our following equilibrium AIMD simulation, indicating such Ca complexes can adsorb and precipitate on the hydrous Al-rich layers and yielding the positively charged C₃A surface (supplementary materials S8 and S9). This phenomenon supports the experimentally observed positive zeta potentials of C₃A [33-36] and further confirms that Ca ions may inhibit C₃A dissolution at high concentrations [33, 36]. Finally, the surface-bound precipitates

(CaOnb(OH)_x(H₂O)_y) containing several liberated OH groups will form an amorphous calcium hydroxide (CH) layer on the Ca leaching surface. Such amorphous layer is kinetically favored as the local coordination environment is enriched with OH groups and benefits the full hydration of Ca ions, despite the adjacent aqueous solution is significantly undersaturated with respect to even crystalline CH [21]. Moreover, this amorphous layer can also act as the precursor and finally transform to AFt or various AFm phase induced by the thermodynamic driving force [37]. The overall reactions recognize the formation of surface alteration layers (SALs) consisting of Al-rich layer and re-participated Ca complexes during the initial hydration processes of C₃A induced by the ICDP mechanism [13] and emphasize the critical roles of incongruent dissolution-triggered Al-rich layers in determining surface activity and interface reactions [36, 38]. When SO₄²⁻ ions are also included in the electrical double layer (EDL) of the partially dissolved C₃A surface, a pronounced inhibition on C₃A dissolution is realized [36]. This effect is achieved by adsorbing SO₄²⁻ ions on the surface through electrostatic attraction with the surface-bound Ca complexes, thereby obstructing the coordination sites that would be otherwise occupied by H⁺ or OH⁻ ions that are known to catalyze the dissolution of alumina [39] and C₃S [7].

Fig. R1. Schematic diagram of the initial hydration of C₃A and implications for cement hydration. The experimental image is from ref. [23].

5. In the SI information, authors have mentioned about the well-established force field models for exploring interactions between water molecules and cement components [12-15]. Authors should provide set of force field parameters used in these calculations with validated molecular models of C₃A. I could not find validated C₃A force field parameters in the references mentioned above.

Reply: We thus supplemented the relevant reference containing the ReaxFF reactive force field parameters for H/O/Ca/Si/Al/S system according to the reviewer's comments (see supplementary material S4).

6. Comparison between ab-initio and all-atom force field model would make sense only when the force field parameters were specifically developed for C₃A mineral and validated with experimental data.

Reply: The ReaxFF reactive force field parameters for the H/O/Ca/Si/Al/S system were originally developed to investigate the mechanical failure modes of ettringite. These parameters were derived through a series of first-principles calculations, and the resulting elastic constants, bulk phase modulus and shear modulus of ettringite align with experimental data, attesting to the accuracy of the force field parameters in simulating Portland cement [40]. Additionally, these ReaxFF parameters have been employed to elucidate the inhibition mechanism of gypsum on C₃A hydration. The outcomes of these simulations are in accordance with high-level quantum chemical (QC) calculations based on MP2/6-31G (d, p) and MP2 (full)/cc-pVTZ basis sets, as well as relevant experimental phenomena [41]. Thus, we compare the DFT-based calculations with the force-field simulations although these force field parameters are not specifically developed for C₃A mineral. It's crucial to note that such a comparison is to provide additional comparisons or verifications for our DFT calculations since the limited availability of experimental data regarding molecular-scale C₃A hydration. Furthermore, this comparison serves to benchmark the corresponding ReaxFF reactive force field parameters, contributing to a deeper understanding of the hydration mechanisms of C₃A based on all-atom models and force-field molecular dynamics

simulations.

7. A minor comment is the necessity to improve the language and grammatical structure of the manuscript.

Reply: We have polished the entire manuscript and improved the language and grammatical structure of the manuscript.

References:

- [1] B.A. Legg, M.D. Baer, J. Chun, G.K. Schenter, S. Huang, Y. Zhang, Y. Min, C.J. Mundy, J.J. De Yoreo, Visualization of Aluminum Ions at the Mica Water Interface Links Hydrolysis State-to-Surface Potential and Particle Adhesion, *J Am Chem Soc*, 142 (2020) 6093-6102.
- [2] M. Azimzadeh Sani, N.G. Pavlopoulos, S. Pezzotti, A. Serva, P. Cignoni, J. Linnemann, M. Salanne, M.P. Gaigeot, K. Tschulik, Unexpectedly High Capacitance of the Metal Nanoparticle/Water Interface: Molecular-Level Insights into the Electrical Double Layer, *Angew Chem Int Ed Engl*, 61 (2022) e202112679.
- [3] Y.H. Wang, S. Zheng, W.M. Yang, R.Y. Zhou, Q.F. He, P. Radjenovic, J.C. Dong, S. Li, J. Zheng, Z.L. Yang, G. Attard, F. Pan, Z.Q. Tian, J.F. Li, In situ Raman spectroscopy reveals the structure and dissociation of interfacial water, *Nature*, 600 (2021) 81-85.
- [4] Q. Chen, J. Liu, B. Yang, Identifying the key steps determining the selectivity of toluene methylation with methanol over HZSM-5, *Nat Commun*, 12 (2021) 3725.
- [5] R. Reocreux, E. Girel, P. Clabaut, A. Tuel, M. Besson, A. Chaumonnot, A. Cabiac, P. Sautet, C. Michel, Reactivity of shape-controlled crystals and metadynamics simulations locate the weak spots of alumina in water, *Nat Commun*, 10 (2019) 3139.
- [6] B. Yoon, G.A. Voth, Elucidating the Molecular Mechanism of CO₂ Capture by Amino Acid Ionic Liquids, *J Am Chem Soc*, 145 (2023) 15663-15667.
- [7] Y. Li, H. Pan, Q. Liu, X. Ming, Z. Li, Ab initio mechanism revealing for tricalcium silicate dissolution, *Nat Commun*, 13 (2022) 1253.
- [8] C.Y. Li, J.B. Le, Y.H. Wang, S. Chen, Z.L. Yang, J.F. Li, J. Cheng, Z.Q. Tian, In situ probing electrified interfacial water structures at atomically flat surfaces, *Nat Mater*, 18 (2019) 697-701.
- [9] S.A. Saslow, S.N. Kerisit, T. Varga, S.T. Mergelsberg, C.L. Corkhill, M.M.V. Snyder, N.M. Avalos, A.S. Yorkshire, D.J. Bailey, J. Crum, R.M. Asmussen, Immobilizing

Pertechnetate in Ettringite via Sulfate Substitution, *Environmental Science & Technology*, 54 (2020) 13610-13618.

[10] A. Kunhi Mohamed, P. Moutzouri, P. Berruyer, B.J. Walder, J. Siramanont, M. Harris, M. Negroni, S.C. Galmarini, S.C. Parker, K.L. Scrivener, L. Emsley, P. Bowen, The Atomic-Level Structure of Cementitious Calcium Aluminate Silicate Hydrate, *J Am Chem Soc*, 142 (2020) 11060-11071.

[11] A. Morales-Melgares, Z. Casar, P. Moutzouri, A. Venkatesh, M. Cordova, A. Kunhi Mohamed, K.L. Scrivener, P. Bowen, L. Emsley, Atomic-Level Structure of Zinc-Modified Cementitious Calcium Silicate Hydrate, *Journal of the American Chemical Society*, 144 (2022) 22915-22924.

[12] S. Ye, P. Feng, Y. Liu, J. Liu, J.W. Bullard, Dissolution and early hydration of tricalcium aluminate in aqueous sulfate solutions, *Cement and Concrete Research*, 137 (2020) 106191.

[13] T. Geisler, L. Dohmen, C. Lenting, M.B.K. Fritzsche, Real-time in situ observations of reaction and transport phenomena during silicate glass corrosion by fluid-cell Raman spectroscopy, *Nature Materials*, 18 (2019) 342-348.

[14] S.S. Lee, P. Fenter, K.L. Nagy, N.C. Sturchio, Real-time observation of cation exchange kinetics and dynamics at the muscovite-water interface, *Nat Commun*, 8 (2017) 15826.

[15] M. Siebecker, W. Li, S. Khalid, D. Sparks, Real-time QEXAFS spectroscopy measures rapid precipitate formation at the mineral-water interface, *Nat Commun*, 5 (2014) 5003.

[16] Y.-C. Zou, L. Mogg, N. Clark, C. Bacaksiz, S. Milovanovic, V. Sreepal, G.-P. Hao, Y.-C. Wang, D.G. Hopkinson, R. Gorbachev, S. Shaw, K.S. Novoselov, R. Raveendran-Nair, F.M. Peeters, M. Lozada-Hidalgo, S.J. Haigh, Ion exchange in atomically thin clays and micas, *Nature Materials*, 20 (2021) 1677-1682.

[17] P. Atkins, J. De Paula, J. Keeler, *Atkins' physical chemistry*, Oxford university press 2023.

[18] A.C. Lasaga, J. Kirkpatrick, *Kinetics of geochemical processes*, Walter de Gruyter GmbH & Co KG 2018.

[19] S. Ye, P. Feng, J. Lu, L. Zhao, Q. Liu, Q. Zhang, J. Liu, J.W. Bullard, Solubility of tricalcium aluminate from 10 °C to 40 °C, *Cement and Concrete Research*, 162 (2022) 106989.

- [20] S. Ye, P. Feng, Y. Liu, J. Liu, J.W. Bullard, In situ nano-scale observation of C₃A dissolution in water, *Cement and Concrete Research*, 132 (2020) 106044.
- [21] A.S. Brand, J.W. Bullard, Dissolution Kinetics of Cubic Tricalcium Aluminate Measured by Digital Holographic Microscopy, *Langmuir*, 33 (2017) 9645-9656.
- [22] A.S. Brand, S.B. Feldman, P.E. Stutzman, A.V. Ievlev, M. Lorenz, D.C. Pagan, S. Nair, J.M. Gorham, J.W. Bullard, Dissolution and initial hydration behavior of tricalcium aluminate in low activity sulfate solutions, *Cement and Concrete Research*, 130 (2020) 105989.
- [23] C. Hesse, F. Goetz-Neunhoeffer, J. Neubauer, A new approach in quantitative in-situ XRD of cement pastes: Correlation of heat flow curves with early hydration reactions, *Cement and Concrete Research*, 41 (2011) 123-128.
- [24] H. Liu, W. Chen, R. Pan, Z. Shan, A. Qiao, J.W.E. Drewitt, L. Hennet, S. Jahn, D.P. Langstaff, G.A. Chass, H. Tao, Y. Yue, G.N. Greaves, From Molten Calcium Aluminates through Phase Transitions to Cement Phases, *Adv Sci*, 7 (2020) 1902209.
- [25] H. Manzano, J.S. Dolado, A. Ayuela, Structural, Mechanical, and Reactivity Properties of Tricalcium Aluminate Using First-Principles Calculations, *Journal of the American Ceramic Society*, 92 (2009) 897-902.
- [26] Y. Tao, S. Zare, F. Wang, M.J.A. Qomi, Atomistic thermodynamics and kinetics of dicalcium silicate dissolution, *Cement and Concrete Research*, 157 (2022) 106833.
- [27] Q. Zheng, C. Liang, J. Jiang, H. Mao, K.C. Bustillo, C. Song, J.A. Reimer, P.J.M. Monteiro, H. Zheng, S. Li, Atomic-scale identification of defects in alite, *Cement and Concrete Research*, 176 (2024) 107391.
- [28] R. Réocreux, T. Jiang, M. Iannuzzi, C. Michel, P. Sautet, Structuration and Dynamics of Interfacial Liquid Water at Hydrated γ -Alumina Determined by ab Initio Molecular Simulations: Implications for Nanoparticle Stability, *ACS Applied Nano Materials*, 1 (2017) 191-199.
- [29] Y. Li, H. Ai, K.H. Lo, Y. Kong, H. Pan, Zongjin Li, Insight into adsorption mechanism of water on tricalcium silicate from first-principles calculations, *Cement and Concrete Research*, 152 (2022) 106684.
- [30] N. Giraud, P.G. Weidler, F. Laye, M. Schwotzer, J. Lahann, C. Wöll, P. Thissen, Corrosion of Concrete by Water-Induced Metal-Proton Exchange, *The Journal of Physical Chemistry C*, 120 (2016) 22455-22459.
- [31] P. Thissen, Exchange Reactions at Mineral Interfaces, *Langmuir*, 36 (2020) 10293-

10306.

[32] C.A. Ohlin, E.M. Villa, J.R. Rustad, W.H. Casey, Dissolution of insulating oxide materials at the molecular scale, *Nat Mater*, 9 (2010) 11-19.

[33] M.E. Tadros, W.Y. Jackson, J. Skalny, Study of the dissolution and electrokinetic behavior of tricalcium aluminate, in: M. Kerker (Ed.) *Hydrosols and Rheology*, Academic Press 1976, pp. 211-223.

[34] S. Joseph, J. Skibsted, Ö. Cizer, A quantitative study of the C₃A hydration, *Cement and Concrete Research*, 115 (2019) 145-159.

[35] K. Yoshioka, E.-i. Tazawa, K. Kawai, T. Enohata, Adsorption characteristics of superplasticizers on cement component minerals, *Cement and Concrete Research*, 32 (2002) 1507-1513.

[36] R.J. Myers, G. Geng, J. Li, E.D. Rodriguez, J. Ha, P. Kidkhunthod, G. Sposito, L.N. Lammers, A.P. Kirchheim, P.J. Monteiro, Role of Adsorption Phenomena in Cubic Tricalcium Aluminate Dissolution, *Langmuir*, 33 (2017) 45-55.

[37] T. Hirsch, T. Matschei, D. Stephan, The hydration of tricalcium aluminate (Ca₃Al₂O₆) in Portland cement-related systems: A review, *Cement and Concrete Research*, 168 (2023) 107150.

[38] R.J. Myers, G. Geng, E.D. Rodriguez, P. da Rosa, A.P. Kirchheim, P.J.M. Monteiro, Solution chemistry of cubic and orthorhombic tricalcium aluminate hydration, *Cement and Concrete Research*, 100 (2017) 176-185.

[39] A. Packter, H.S. Dhillon, Kinetics and mechanism of the heterogeneous reactions of γ -, χ -, and α - aluminas with aqueous sodium hydroxide solutions, *Journal of the Chemical Society A: Inorganic, Physical, Theoretical*, (1970) 1266-1270.

[40] L. Liu, A. Jaramillo-Botero, W.A. Goddard, 3rd, H. Sun, Development of a ReaxFF reactive force field for ettringite and study of its mechanical failure modes from reactive dynamics simulations, *J Phys Chem A*, 116 (2012) 3918-3925.

[41] J. Zhu, D. Shen, W. Wu, B. Jin, S. Wu, Hydration inhibition mechanism of gypsum on tricalcium aluminate from ReaxFF molecular dynamics simulation and quantum chemical calculation, *Molecular Simulation*, 47 (2021) 1465-1476.

Reviewer #1 (Remarks to the Author):

The authors have addressed the comments. Therefore, the paper may be accepted for publication

I have also been asked to comment of the response to reviewer #3's comments. From my perspective, the authors have addressed reviewer #3's comments adequately.

Reviewer #2 (Remarks to the Author):

The article is interesting as it reveals the initial hydration progress of tricalcium aluminate (C3A) at the molecular scale. The authors have given nice and detailed responses to the comments and have revised the article based on these comments. After the revision, the article is clearer compared with the one in the original manuscript. The Introduction was also rephrased. Overall, the article has improved a lot after the revision. Therefore, my suggestion is to accept the article.